# EVC: Towards Real-Time Neural Image Compression with Mask Decay

**Guo-Hua Wang**[1]\*, **Jiahao Li**[2], **Bin Li**[2], **Yan Lu**[2]
[1]State Key Laboratory for Novel Software Technology, Nanjing University
[2]Microsoft Research Asia
`wangguohua@lamda.nju.edu.cn, {li.jiahao,libin,yanlu}@microsoft.com`

## Abstract

Neural image compression has surpassed state-of-the-art traditional codecs (H.266/VVC) for rate-distortion (RD) performance, but suffers from large complexity and separate models for different rate-distortion trade-offs. In this paper, we propose an Efficient single-model Variable-bit-rate Codec (EVC), which is able to run at 30 FPS with 768x512 input images and still outperforms VVC for the RD performance. By further reducing both encoder and decoder complexities, our small model even achieves 30 FPS with 1920x1080 input images. To bridge the performance gap between our different capacities models, we meticulously design the mask decay, which transforms the large model's parameters into the small model automatically. And a novel sparsity regularization loss is proposed to mitigate shortcomings of $L_p$ regularization. Our algorithm significantly narrows the performance gap by 50% and 30% for our medium and small models, respectively. At last, we advocate the scalable encoder for neural image compression. The encoding complexity is dynamic to meet different latency requirements. We propose decaying the large encoder multiple times to reduce the residual representation progressively. Both mask decay and residual representation learning greatly improve the RD performance of our scalable encoder. Our code is at `https://github.com/microsoft/DCVC`.

## 1 Introduction

The image compression based on deep learning has achieved extraordinary rate-distortion (RD) performance compared to traditional codecs (H.266/VVC) (Bross et al., 2021). However, two main issues limit its practicability in real-world applications. One is the large complexity. Most state-of-the-art (SOTA) neural image codecs rely on complex models, such as the large capacity backbones (Zhu et al., 2022; Zou et al., 2022), the sophisticated probability model (Cheng et al., 2020), and the parallelization-unfriendly auto-regressive model (Minnen et al., 2018). The large complexity easily results in unsatisfied latency in real-world applications. The second issue is the inefficient rate-control. Multiple models need to be trained and stored for different rate-distortion trade-offs. For the inference, it requires loading specific model according to the target quality. For practical purposes, a single model is desired to handle variable RD trade-offs.

In this paper, we try to solve both issues to make a single real-time neural image compression model, while its RD performance is still on-par with that of other SOTA models. Inspired by recent progress, we design an efficient framework that is equipped with Depth-Conv blocks (Liu et al., 2022) and the spatial prior (He et al., 2021; Li et al., 2022a; Qian et al., 2022). For variable RD trade-offs, we introduce an adjustable quantization step (Chen & Ma, 2020; Li et al., 2022a) for the representations. All modules within our framework are highly efficient and GPU friendly, different from recent Transformer based models (Zou et al., 2022). Encoding and decoding (including the arithmetic coding) achieves 30 FPS for the $768 \times 512$ inputs. Compared with other SOTA models, ours enjoys comparable RD performance, low latency, and a single model for all RD trade-offs.

To further accelerate our model, we reduce the complexity of both the encoder and decoder. Three series models are proposed: Large, Medium, and Small (cf. Appendix Tab. 3). Note that our small

---

\*This work was done when Guo-Hua Wang was a full time intern at Microsoft Research Asia.

Figure 1: The trade-off between BD-Rate and complexities on Kodak. The anchor is VTM. (a) and (b) show the performance improvement by our mask decay and residual representation learning (RRL). We cite results from SwinT-Hyperprior (Zhu et al., 2022) for comparison.

model even achieves 30 FPS for $1920 \times 1080$ inputs. Recent work (Zhu et al., 2022) points out that simply reducing the model's capacity results in significant performance loss. However, this crucial problem has not been solved effectively. In this paper, we advocate training small neural compression models with a teacher to mitigate this problem. In particular, *mask decay* is proposed to transform the large model's parameters into the small model automatically. Specifically, mask layers are first inserted into a pretrained teacher model. Then we sparsify these masks until all layers become the student's structure. Finally, masks are merged into the raw layers meanwhile keeping the network functionality unchanged. For the sparsity regularization, $L_p$-norm based losses are adopted by most previous works (Li et al., 2016; Ding et al., 2018; Zhang et al., 2021). But they hardly work for neural image compression (cf. Fig. 7). In this paper, we propose a novel sparsity regularization loss to alleviate the drawbacks of $L_p$ so that optimize our masks more effectively. With the help of our large model, our medium and small models are improved significantly by 50% and 30%, respectively (cf. Fig. 1a). That demonstrates the effectiveness of our mask decay and reusing a large model's parameters is helpful for training a small neural image compression model.

In addition, considering the various device capabilities in real-world codec applications, the encoder scalability of supporting different encoding complexities while with only one decoder is also a critical need. To achieve this, we propose compressing the cumbersome encoder multi-times to progressively bridge the performance gap. Both the residual representation learning (RRL) and mask decay treat the cumbersome encoder as a reference implicitly, and encourage the diversity of different small encoders. Therefore ours achieves superior performance than training separate encoders (cf. Fig. 1b). And compared to SlimCAE (Yang et al., 2021) which is inspired by slimmable networks (Yu et al., 2019), ours enjoys a simple framework and better RD performance.

Our contributions are as follows.

- We propose an Efficient Variable-bit-rate Codec (EVC) for image compression. It enjoys only one model for different RD trade-offs. Our model is able to run at 30 FPS for the $768 \times 512$ inputs, while is on-par with other SOTA models for the RD performance. Our small model even achieves 30 FPS for the $1920 \times 1080$ inputs.
- We propose mask decay, an effective method to improve the student image compression model with the help of the teacher. A novel sparsity regularization loss is also introduced, which alleviates the shortcomings of $L_p$ regularization. Thanks to mask decay, our medium and small models are significantly improved by 50% and 30%, respectively.
- We enable the encoding scalability for neural image compression. With residual representation learning and mask decay, our scalable encoder significantly narrows the performance gap from the teacher and achieves a superior RD performance than previous SlimCAE.

## 2 RELATED WORKS

**Neural image compression** is in a scene of prosperity. Ballé et al. (2017) proposes replacing the quantizer with an additive i.i.d. uniform noise, so that the neural image compression model enjoys the end-to-end training. Then, the hyperprior structure was proposed by Ballé et al. (2018).

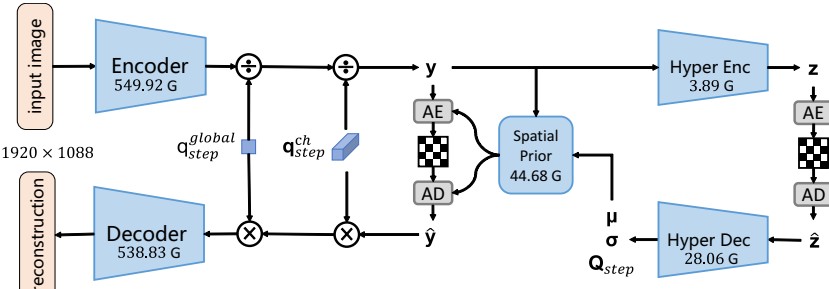

Figure 2: The overall framework of EVC.

After that, numerous methods are proposed to improve the entropy model and the backbone (both encoder and decoder). Auto-regressive component (Minnen et al., 2018) and Gaussian Mixture Model (Cheng et al., 2020) are proposed to improve the probability estimation in entropy model. Informer (Kim et al., 2022) proposes using an attention mechanism to exploit both global and local information. Recently, more powerful backbones (e.g., INNs (Xie et al., 2021), Transformer (Qian et al., 2022) and Swin (Zhu et al., 2022)) are introduced to replace ResNet (He et al., 2016) in the encoder and decoder. However, recent SOTA neural image codec models suffer from high latency and separate models for each RD trade-off. In this paper, we propose an efficient neural image compression framework that aims at encoding and decoding in real-time. Especially, our framework can also handle variable rate-distortion trade-offs by the adjustable quantization step (Chen & Ma, 2020; Cui et al., 2021; Li et al., 2022a). The RD performance of our models is on-par with other SOTA neural image compression models, and surpasses the SOTA traditional image codec (H.266/VVC).

**Model compression and knowledge distillation (KD)** are two prevalent techniques for accelerating neural networks. Model compression, especially filter level pruning (He et al., 2017), aims at obtaining an efficient network structure from a cumbersome network. Some works (Li et al., 2016; Luo & Wu, 2020; He et al., 2019; Molchanov et al., 2019) calculate the importance of each filter, and prune the unimportant filters. Other methods directly penalize filters during training. They treat some parameters as the importance implicitly (Liu et al., 2017; Ding et al., 2021; Wang & Wu, 2022). $L_p$-norm based sparsity losses are used in these methods (Ding et al., 2018; Wen et al., 2016; Zhang et al., 2021). For a pruned network, KD (Hinton et al., 2015) is utilized to boost the performance by mimicking the teacher's knowledge (e.g., the logit (Zhao et al., 2022), or the feature (Wang et al., 2022)). All these techniques are mainly studied on semantic tasks, but rarely advocated for neural image compression. This paper firstly tries to improve a small neural image compression model with the help of a large model. And we claim that, for this task, reusing the teacher's filters is an effective way to improve the student. To achieve this, we propose mask decay with a novel sparsity regularization loss. With all proposed techniques, our student models are greatly improved.

## 3    METHODOLOGY

We first introduce our efficient framework briefly. Then, three models with different complexities are proposed: Large, Medium, and Small. Next, we describe how to improve our small model (student) with the help of our large model (teacher). Mask decay is proposed to transform the teacher's parameters into the student automatically. At last, the scalable encoder is advocated. With residual representation learning and mask decay, several encoders are trained progressively to bridge the performance gap from the teacher.

### 3.1    OUR EVC FOR IMAGE COMPRESSION

The framework of EVC is shown in Fig. 2. Given an image, the encoder generates its representation $\boldsymbol{y}$. Following Chen & Ma (2020); Li et al. (2022a), to achieve variable bit-rates within a single model, we introduce adjustable quantization steps. $q_{\text{step}}^{\text{global}}$ and $\boldsymbol{q}_{\text{step}}^{\text{ch}}$ are two learnable parameters that quantifying $\boldsymbol{y}$ globally and channel-wisely, respectively. And different RD trade-offs can be achieved by adjusting the magnitude of $q_{\text{step}}^{\text{global}}$. More details are introduced in Appendix Sec. A.

Fig. 2 also presents MACs of each module with the $1920 \times 1088$ input image. Obviously, most computational costs come from the encoder and the decoder. Hence, we focus on accelerating these

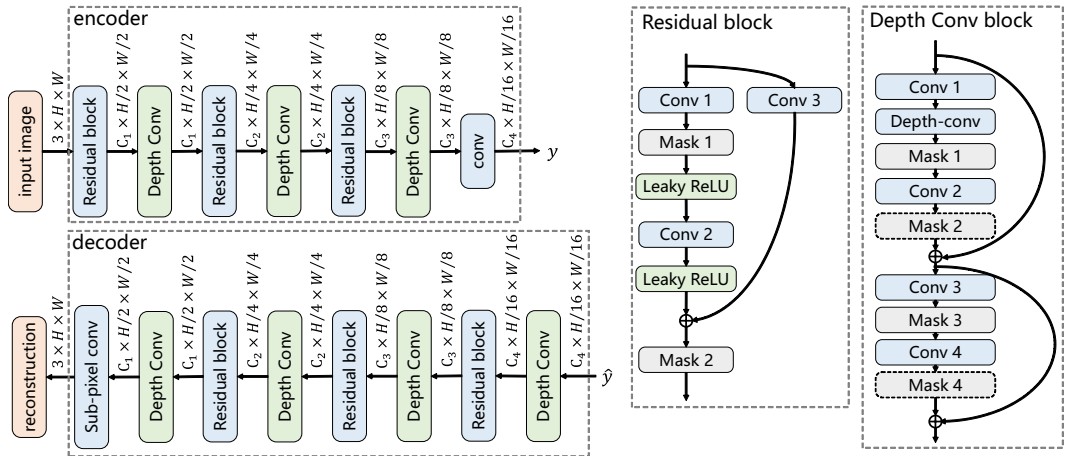

Figure 3: The architectures of our encoder and decoder. All mask layers will be merged into conv. layers after training. For simplification, we omit Leaky ReLUs within the Depth-Conv block.

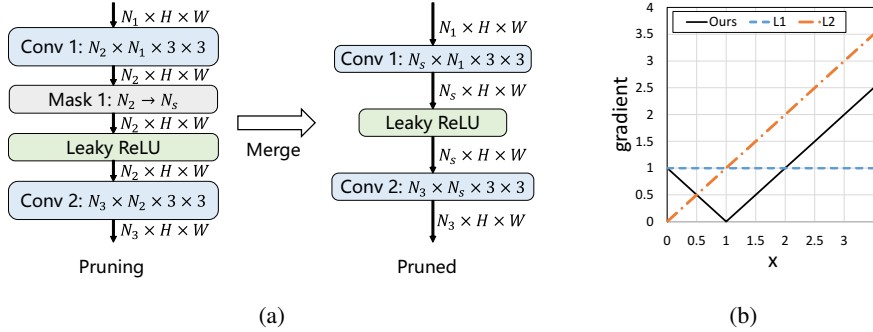

(a)        (b)

Figure 4: (a) presents our pruning method. The mask layer is inserted between two conv. layers. These two cumbersome conv. layers will be transformed into efficient ones by merging the mask layer into them. (b) compares the gradient $\frac{\partial \mathcal{L}_{sparse}}{\partial x}$ of different sparsity loss functions.

two modules, while keeping other parts' architectures unchanged. Fig. 3 shows the detail structure of these two modules. Different from recent SOTA models (Zhu et al., 2022; Qian et al., 2022; Zou et al., 2022), we adopt residual blocks and depth-wise convolution blocks (Liu et al., 2022) rather than Transformer. Hence, ours is more GPU-friendly and enjoys lower latency. $C_1$, $C_2$, $C_3$, and $C_4$ are pre-defined numbers for different computational budgets. Three models are proposed with channels $[64, 64, 128, 192]$ (Small), $[128, 128, 192, 192]$ (Medium), and $[192, 192, 192, 192]$ (Large). Tab. 3 in the appendix summarizes models' #Params and MACs. Note that Large is our cumbersome teacher model while Medium and Small are efficient student models.

## 3.2 IMPROVE THE STUDENT BY MASK DECAY

Due to the limited capacity, the student performs badly if training from scratch (Zhu et al., 2022). How to improve the student is a crucial but rarely studied problem for neural image compression. In this paper, we claim that reusing the teacher's parameters is helpful. To achieve it, we propose mask decay to transform parameters in an end-to-end manner.

**Compress one layer's representation by mask decay.** As showed in Fig. 4a, a mask layer is inserted between two layers to reduce the representation's channels (from $N_2$ to $N_s$). This mask layer contains $N_2$ parameters to prune corresponding channels. In the forward computation, the representation multiplies these parameters channel-wisely. During training, these parameters will become sparse by optimizing our sparsity loss. After training, indicated by the mask, useless parameters in Conv #1 and #2 are pruned. And the mask is eliminated by multiplying itself into Conv #2.

To sparsify parameters $\boldsymbol{m}$ in the mask layer, we design a novel sparsity regularization loss. Most previous works (Li et al., 2016; Wen et al., 2016; Zhang et al., 2021) adopt L1-norm ($|\boldsymbol{m}|$) or L2-norm ($\frac{1}{2}\|\boldsymbol{m}\|_2$) based functions. Note that with the gradient descend, $\boldsymbol{m}$ is updated by $sign(\boldsymbol{m})$ and

Figure 5: Illustration of compressing encoders multi-times to learn residual representations progressively. "Enc", "Dec", and "E" denote the encoder, the decoder, and the entropy module, respectively.

$m$, respectively. Miserably, both of them fail for neural image compression models (cf. Fig. 7). We argue that, for a parameter with a positive value, the gradient derived by the L1-norm is a constant without considering its own magnitude. And for the L2-norm, when the parameter is approaching zero, the gradient is too small to sparsify the mask. In addition, both of them have no stationary points when parameters are positive. To alleviate these problems, We design a novel sparsity regularization loss from the gradient view. As showed in Fig. 4b, the gradient is defined by

$$\frac{\partial \mathcal{L}_{sparse}(x)}{\partial x} = |x - 1| \,, \tag{1}$$

where $x \geq 0$. For $x > 1$, the parameter with a large magnitude suffers a large gradient to decay, which likes the L2-norm. And for $0 \leq x < 1$, a smaller parameter enjoys a larger gradient to approach zero fast. In addition, there is a stationary point at $x = 1$, which means $\frac{\partial \mathcal{L}_{sparse}(x)}{\partial x}|_{x=1} = 0$. When it comes to L1 and L2, there always is a positive gradient for $x > 0$. To make the model converge to stable, it needs a negative gradient from the task loss to balance this positive gradient. Thanks to our sparsity loss, only a small negative gradient is needed when $x$ is close to 1. $\mathcal{L}_{sparse}$ can be calculated by integrating Equation 1, which is

$$\mathcal{L}_{sparse}(x) = \begin{cases} -\frac{1}{2}x^2 + x, & \text{if } 0 \leq x \leq 1 \,, \\ \frac{1}{2}x^2 - x + 1, & \text{if } x > 1 \,. \end{cases} \tag{2}$$

For training, inspired by AdamW (Loshchilov & Hutter, 2018), we decouple $\mathcal{L}_{sparse}$ from the task loss $\mathcal{L}_{RD}$ as weight decay. Therefore, the task loss is not affected by our sparsity loss explicitly. So we name our algorithm as *mask decay*. For the mask $m$, the updating formula is

$$m_{t+1} = m_t - \eta \frac{\partial \mathcal{L}_{sparse}}{\partial m_t} - \gamma \frac{\partial \mathcal{L}_{RD}}{\partial m_t} = m_t - \eta |m_t - 1| - \gamma \frac{\partial \mathcal{L}_{RD}}{\partial m_t} \,, \tag{3}$$

where $\gamma$ is the learning rate and $\eta$ is the rate for mask decay.

**Mask decay for our encoder and decoder.** As illustrated in Fig. 3, to transform the teacher into the student, we insert mask layers in some positions if necessary. Our principle is to insert mask layers as few as possible. In one residual block, Mask #1 and #2 are inserted to prune channels inside and outside the residual connection, respectively. In the depth-conv block, Mask #1 determines channels of the depth convolution layer, while Mask #3 prunes representations between Conv #3 and #4. Note that the output channels of these two blocks are shared, so Mask #2 and #4 in the depth-conv block share weights with Mask #2 in the residual block. And we place Mask #2 and #4 behind the residual adding to guarantee the identity gradient back-propagation.

**Training with mask decay.** The training begins with a pretrained teacher model. Then, we insert mask layers and initialize them with **1**. The whole network is optimized by minimizing $\mathcal{L}_{RD} = R + \lambda D$, where $\lambda$ steers the trade-off between the rate $R$ and the distortion $D$. Our masks are decayed before each iteration, likely weight decay (Loshchilov & Hutter, 2018). During training, we stop mask decay for one mask if it is sparse enough. When all masks meet the sparsity requirements, the whole sparse network can transform into the student by merging masks, meanwhile keeping the network functionality unchanged. At last, the student will be finetuned for a few epochs.

### 3.3 THE SCALABLE ENCODER

At last, we advocate the scalable encoder in neural image codecs. A trivial solution for the scalable encoder is directly training separate small encoders with a uniform decoder. For a specific encoding

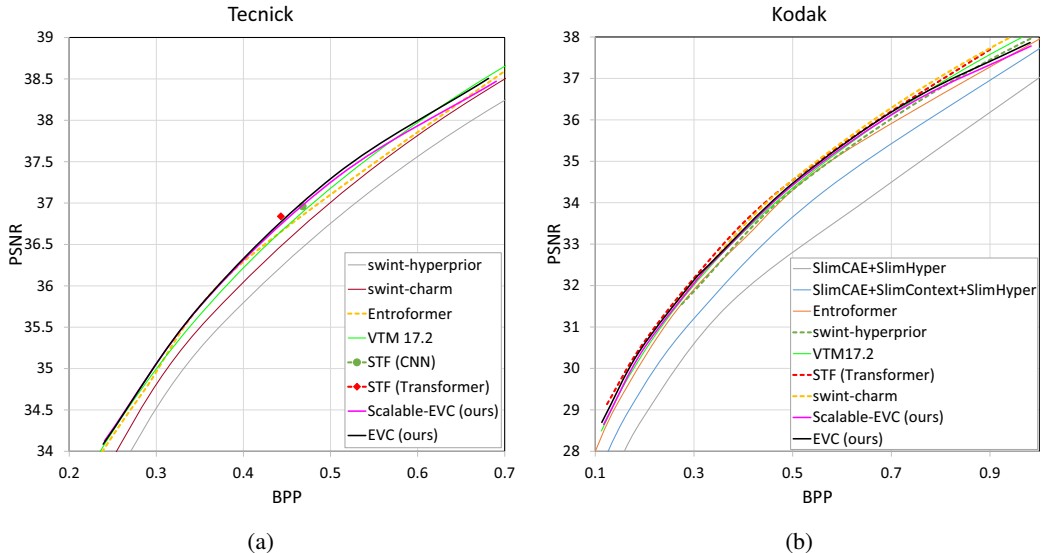

(a)                                           (b)

Figure 6: Rate Distortion curves for Tecnick and Kodak. Our models are on-par with swint (Zhu et al., 2022), Entroformer (Qian et al., 2022), STF (Zou et al., 2022), and outperforms the traditional method VTM. Note that our models are dramatically faster than these methods (cf. Tab. 1).

complexity, the corresponding number of encoders are chosen to encode together, and then the best bitstreams will be sent to the decoder. However, these separate encoders suffer from the homogenization problem. We propose residual representation learning (RRL) to mitigate this problem. As illustrated in Fig. 5, we compress the cumbersome encoder several times to learn the residual representations progressively. In each step, only parameters in the compressing encoder are optimized. That makes sure previous encoders generate decodable bitstreams. For a specific encoding complexity, the first $k$ encoders are applied. Our RRL encourages the encoders' diversity. Both RRL and mask decay treat the cumbersome teacher as a reference, which makes the training more effective. SlimCAE (Yang et al., 2021) uses slimmable layers (Yu & Huang, 2019) to obtain an image compression model with dynamic complexities. Our framework is more simple to implement and achieves superior RD performance. CBANet (Guo et al., 2021) proposes a multi-branch structure for dynamic complexity. It can be considered as our baseline. More details are in Appendix C.

## 4 EXPERIMENTS

**Training & Testing.** The training dataset is the training part of Vimeo-90K (Xue et al., 2019) septuplet dataset. For a fair comparison, all models are trained with 200 epochs. For our method, it costs 60 epochs for the teacher to transform into the student with mask decay, then the student is finetuned by 140 epochs. Testing datasets contain Kodak (Franzen, 1999), HEVC test sequences (Sharman & Suehring, 2017), Tecnick (Asuni & Giachetti, 2014). BD-Rate (Bjontegaard, 2001) for peak signal-to-noise ratio (PSNR) versus bits-per-pixel (BPP) is our main metric to compare different models. More details about experimental settings are introduced in Appendix Sec. D.

### 4.1 COMPARISON WITH STATE-OF-THE-ART

Fig. 6 compares our models with other SOTA models on Tecnick and Kodak. Our large model ("EVC (ours)") outperforms the SOTA traditional image codec ("VTM 17.2") and is on-par with other SOTA neural image codec models. Note that EVC handles different RD trade-offs by only *one* model, while other SOTA methods train separate models for each trade-off. Tab. 1 compares latency of different models. For Entroformer (Qian et al., 2022) and STF (Zou et al., 2022), we measure the latency by running their official code. Note that the latency *contains arithmetic coding*, but ignores the disk I/O time. SwinT (Zhu et al., 2022) is also compared in Appendix Tab. 6. Our models surpass these SOTA models with a significant margin. Our large model runs at 30 FPS on A100 for the $768 \times 512$ inputs, while our small model even achieves 30 FPS for the 1080P inputs. For more previous but high-impact neural image codecs (e.g., Cheng2020 (Cheng et al., 2020) and Xie2021 (Xie et al., 2021)), we make comparisons in Appendix F.

Table 1: Latency (ms) comparison. Lower means better. Entroformer (Qian et al., 2022) and STF (Zou et al., 2022) are the SOTA methods. 'OM' means out of memory. Note that 2080Ti and A100 have 12GB and 80GB memory, respectively. **Bold** denotes running over 30 FPS.

| Resolution | GPU | Type | Entroformer | STF | | EVC | | |
| | | | | Transformer | CNN | Large | Medium | Small |
|---|---|---|---|---|---|---|---|---|
| $768 \times 512$ | 2080Ti | encoding | OM | 176.3 | 158.5 | 63.0 | 44.7 | **28.4** |
| | | decoding | OM | 202.3 | 210.2 | 41.1 | **32.4** | **24.4** |
| | A100 | encoding | 816.8 | 115.9 | 96.4 | **21.1** | **19.8** | **17.7** |
| | | decoding | 4361.9 | 143.2 | 118.0 | **19.1** | **17.1** | **15.6** |
| $1920 \times 1080$ | 2080Ti | encoding | OM | 576.0 | 456.0 | 305.3 | 181.5 | 90.9 |
| | | decoding | OM | 531.7 | 652.0 | 179.2 | 118.1 | 73.2 |
| | A100 | encoding | 7757.4 | 355.6 | 278.1 | 84.2 | 56.3 | **31.4** |
| | | decoding | OM | 354.8 | 281.7 | 60.2 | 46.5 | **29.7** |

Table 2: BD-Rate (%) comparison for PSNR. L, M, and S denote Large, Medium, and Small, respectively. The anchor is VTM. Lower BD-Rate means better. MACs is for $1920 \times 1088$ inputs.

| Enc | Dec | MACs (G) | Method | HEVC | | | | | Kodak |
| | | | | B | C | D | E | Avg. | |
|---|---|---|---|---|---|---|---|---|---|
| L | L | 1165.39 | Baseline | -5.2 | -0.9 | -0.9 | -3.5 | -2.63 | -1.1 |
| M | L | 878.37 | Baseline | -2.8 | 2.2 | 2.5 | 0.0 | 0.48 | 1.1 |
| | | | Mask Decay | **-4.5** | **-0.2** | **0.2** | **-3.0** | **-1.88 (76%↑)** | **-0.4 (68%↑)** |
| S | L | 688.64 | Baseline | 0.3 | 7.1 | 7.0 | 2.8 | 4.30 | 4.0 |
| | | | Mask Decay | **-1.9** | **2.9** | **3.5** | **0.3** | **1.20 (45%↑)** | **1.6 (47%↑)** |
| L | M | 885.12 | Baseline | -2.1 | 3.1 | 3.6 | 1.1 | 1.43 | 2.1 |
| | | | Mask Decay | **-4.0** | **0.7** | **0.8** | **-1.6** | **-1.03 (60%↑)** | **0.2 (59%↑)** |
| L | S | 700.38 | Baseline | 2.1 | 9.5 | 10.7 | 7.7 | 7.50 | 7.0 |
| | | | Mask Decay | **-0.4** | **5.6** | **6.9** | **3.3** | **3.85 (36%↑)** | **4.0 (37%↑)** |
| M | M | 598.10 | Baseline | -0.8 | 4.7 | 5.0 | 1.6 | 2.63 | 2.9 |
| | | | Mask Decay | **-3.1** | **1.6** | **1.5** | **-0.8** | **-0.20 (54%↑)** | **0.9 (50%↑)** |
| S | S | 223.63 | Baseline | 6.3 | 14.7 | 15.6 | 10.6 | 11.80 | 10.0 |
| | | | Mask Decay | **2.6** | **9.5** | **10.6** | **6.5** | **7.30 (31%↑)** | **6.9 (28%↑)** |

## 4.2 MASK DECAY

Previous work (Zhu et al., 2022) finds small models suffer from poor RD performance. This problem is alleviated by our mask decay. We accelerate the encoder and the decoder respectively to study more cases. Note that the large model is the teacher and the decay rate is $4e-5$. Fig. 1a and Tab. 2 summarize experimental results. In Tab. 2, we also calculate the relative improvement as a percentage. For example, if BD-Rate of the baseline student, our student, and the baseline teacher are $1.1\%$, $-0.4\%$, and $-1.1\%$, respectively. The relative improvement is $\frac{1.1-(-0.4)}{1.1-(-1.1)} = 68\%$. This metric indicates how much the performance gap between the student and the teacher our method narrows. RD curves are shown in Appendix Fig. 17. From experimental results, we conclude that:

- Students are improved by mask decay with a significant margin. For medium and small models, relative improvements are about $50\%$ and $30\%$, respectively. This demonstrates the effectiveness of our method and the student benefits from reusing the teacher's parameters. Compared with SwinT-Hyperprior in Fig. 1a, our EVC achieves a superior trade-off between the RD performance and the complexity. In addition, our models enjoy small sizes.
- The encoder is more redundant than the decoder. Note that our encoder and decoder are symmetric. We surprisingly find compressing the encoder enjoys less RD performance loss, compared with the decoder. With the large decoder, compressing the encoder from Large to Small loses $2.7\%$ BD-Rate on Kodak. On the other hand, only compressing the decoder loses $5.1\%$ BD-Rate.

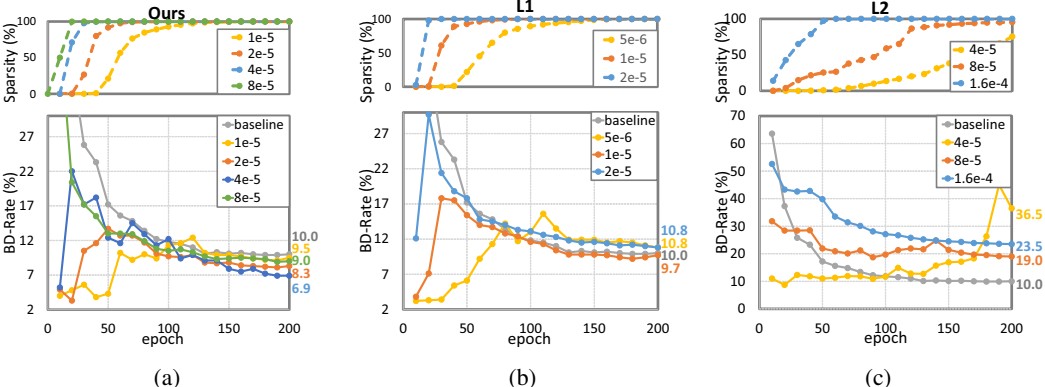

Figure 7: Ablation studies for different sparsity losses. We adjust the decay rate $\eta$ for each criterion. This figure is best viewed in color and zoomed in.

Appendix Fig. 20 visualizes our models' reconstructions. Reconstructions of different capacities models are consistent. Mask decay does not result in extra obvious artifacts. And our encoding and decoding time are dramatically shorter than VTM's.

### 4.3 THE SCALABLE ENCODER

Fig. 1b presents results for the scalable encoder. First, training separate encoders suffers from poor performance, which is due to the homogenization problem. Then, with our framework, training these encoders one-by-one to bridge the residual representations greatly improves the performance. Our RRL encourages these encoders' diversity. Finally, the mask decay treats the cumbersome encoder as a reference and makes training more efficient. More details can be found in Appendix Sec. C.

Fig. 6 shows RD curves for our Scalable-EVC. Ours outperforms SlimCAE (Yang et al., 2021) significantly by $15.5\%$ BD-Rate. Note that SlimCAE is a powerful model equipped with hyperprior (Ballé et al., 2018), conditional convolutions (Choi et al., 2019), and an autoregressive context model (Minnen et al., 2018). Overall, Scalable-EVC is on-par other SOTA models.

### 4.4 ABLATION STUDIES

**Sparsity losses and decay rates.** Fig. 7 summarizes experimental results for mask decay with different sparsity regularization losses and decay rates. For sparsity, $0\%$ and $100\%$ denote the teacher's and the student's structures, respectively. We adjust the decay rate $\eta$ for each sparsity loss to get reasonable sparsity decay curves. BD-Rate is calculated over VTM on Kodak. For our sparsity loss, all decay rates achieve better RD performance compared with the baseline. A suitable decay rate ($4e-5$) achieves the best BD-Rate. A large decay rate ($8e-5$) results in fast decay which destroys the model's performance in early epochs. This model suffers from the insufficient decay process, and the performance is easily saturated. Small decay rates ($1e-5$ and $2e-5$) waste too many epochs for mask decay and the model lacks finetuning. For L1, the model suffers from poor performance with both large and small decay rates. With a suitable rate ($1e-5$), it is still worse than ours with a significant margin. For L2, we find it hardly works because the gradient vanishes when the mask is close to zero.

**Discussions for important channels.** In the field of model pruning, many works (Li et al., 2016; He et al., 2019; Liebenwein et al., 2019; Molchanov et al., 2019) study how to identify unimportant channels for pruning. However, recent works (Liu et al., 2018; Li et al., 2022b) indicate randomly pruning channels results in comparable or even superior performance, which demonstrates there may be no importance difference among all channels. Previous works study mainly semantic tasks. We want to extend this conclusion for low-level tasks, i.e., image compression.

First, we notice that our mask decay selects similar channels with different decay rates in Figure 7a. We use $\mathbb{C}_i$ to represent the set of chosen channels, where $i \in F = \{1e-5, 2e-5, 4e-5, 8e-5\}$ means the decay rate $\eta$. We check masks in the first two blocks, and #channels of the teacher and the student are 192 and 64, respectively. Hence, about $33\%$ channels need to be kept after pruning. However, we find $\cap_{i \in F}\mathbb{C}_i$ and $\cup_{i \in F}\mathbb{C}_i$ contain $23\%$ and $43\%$ channels, respectively. Are these

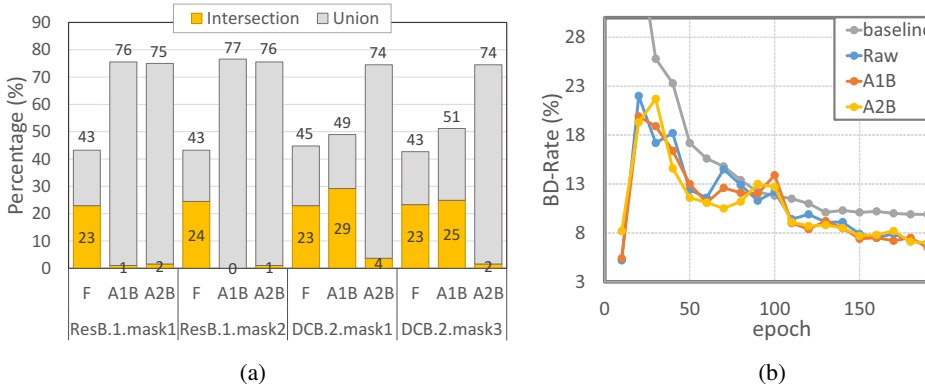

(a)                                                    (b)

Figure 8: (a) presents the percentage of channels w.r.t different mask decay manners in different layers. F, A1B, and A2B mean all channels are free, avoiding the first block, and avoiding the first 2 blocks, respectively. Numbers of the intersection of A$k$B ($k = 1, 2$) and F are small, which means our AMD indeed avoids channels in F. (b) presents the BD-Rate over VTM on Kodak w.r.t training epochs. Raw denotes the curve for $\eta = 4e - 5$ in Figure 7a.

chosen channels important? Our answer is negative. To verify our opinion, we propose mask decay with avoiding specific channels. We divide all channels into two groups: freedom $\mathbb{F}$ and avoidance $\mathbb{A}$. To avoid channels in $\mathbb{A}$, we set a large decay rate for them. Therefore, the loss of our *Avoidable Mask Decay* (AMD) is

$$\mathcal{L}_{AMD}(x) = \eta_f \mathbf{1}_{x \in \mathbb{F}} \mathcal{L}_{sparse}(x) + \eta_a \mathbf{1}_{x \in \mathbb{A}} \mathcal{L}_{sparse}(x) \tag{4}$$

where $\mathbf{1}_{condition}$ is 1 if the condition is true, 0 otherwise. Defaultly, we set $\eta_a = 10\eta_f$. Hence, avoidable channels enjoy a large decay rate to approach zero quickly. We conduct two experiments that avoid selecting channels in $\cup_{i \in F} \mathbb{C}_i$ for the first one or two blocks, respectively. Figure 8a illustrates our AMD indeed avoid channels. Specifically, we calculate $\mathbb{C}_{AkB} \cap \mathbb{C}_F$ and $\mathbb{C}_{AkB} \cup \mathbb{C}_F$, where $k = 1, 2$ denotes avoiding channels in the first $k$ blocks and $\mathbb{C}_F = \cup_{i \in F} \mathbb{C}_i$. At most 1% channels are chosen by both A1D and F in the first block, and A2B avoids channels of F in both two blocks. Figure 8b shows the BD-Rate for each epoch. From it, we find AMD results in comparable or a little bit superior performance compared to the raw mask decay.

Overall, if some channels are important, we can avoid these channels to achieve comparable performance. That demonstrates there are no important channels. For future works, this conclusion guides us to pay more attention to the optimization process rather than the importance of each channel.

## 5    CONCLUSIONS AND FUTURE WORKS

We proposed EVC for image compression. With $768 \times 512$ inputs, EVC is able to run at 30 FPS, while the RD performance is on-par with SOTA neural codecs and superior to the SOTA traditional codec (H.266/VVC). With an adjustable quantization step, EVC handles variable RD trade-off within only one model. These three signs of progress make a new milestone in the field of image codec. By reducing the complexities of both encoder and decoder, a small model is proposed, which achieves 30 FPS for 1080P inputs. We proposed mask decay for training this small model with the help of the cumbersome model. And a novel sparsity regularization loss is designed to alleviate the drawbacks of L1 and L2 losses. Thanks to our algorithm, our medium and small models are improved significantly by 50% and 30%, respectively. At last, the scalable encoder is advocated. With only one decoder, there are several encoders for different complexities. With residual representation learning and mask decay, the scalable encoder narrows the performance gap between the teacher. And our scalable EVC outperforms previous SlimCAE by a significant margin.

We believe this paper will inspire many future works. First, we point out the encoder is more redundant than the decoder. Hence, improving the decoder is promising. Second, this paper mainly focuses on the pre-defined student structure. How to determine the complexity of each layer is still an open problem for image compression models. Third, it is also encouraged to find a more powerful pruning algorithm. But according to our AMD, we advise paying more attention to the optimization process rather than the importance criterion for each filter.

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

# EVC: Towards Real-Time Neural Image Compression with Mask Decay
## -Appendix-

## A  THE FRAMEWORK OF EVC

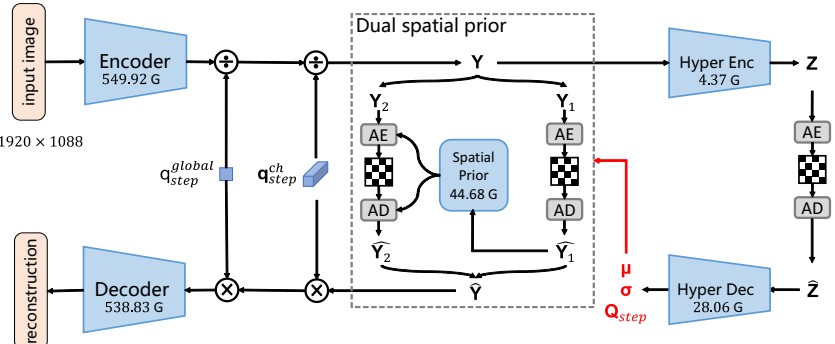

Figure 9: The overall framework of EVC.

Our EVC for image compression is illustrated in Fig. 9. Next, we introduce each module in detail.

**The encoder and the decoder.** The structures of these two modules are illustrated in Fig. 3. The block output dimensionalities are predefined by $C_1$, $C_2$, $C_3$, and $C_4$, respectively. Downsampling is implemented by setting the stride as 2 in specific conv. layers, while upsampling is achieved by the sub-pixel convolution (Shi et al., 2016). In each residual block, the kernel size of both Conv #1 and #2 is $3 \times 3$, while Conv #3's is $1 \times 1$. And Conv #1 enlarges the channel number to the output dimensionality. In each depth-conv block, only the depth conv. layer uses $3 \times 3$ conv., while others adopt $1 \times 1$ conv. Conv #3 enlarges the channels by 4 times, while Conv #4 converts it back.

**The hyper encoder and the hyper decoder.** Fig. 10 describes the structures of our hyperprior. Note that we use $3 \times 3$ conv. in the hyper encoder.

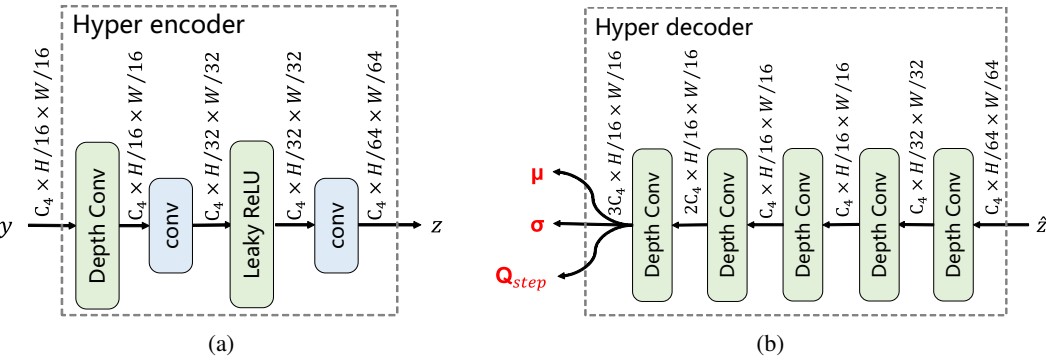

Figure 10: (a) and (b) present the structures of our hyper encoder and hyper decoder, respectively.

**The dual spatial prior.** Following He et al. (2021); Li et al. (2022a); Qian et al. (2022), we adopt dual spatial prior to reduce the spatial redundancy. The detail structure is illustrated in Fig. 11. The representation $\mathbf{Y}$ is split into two parts $\mathbf{Y}_1$ and $\mathbf{Y}_2$ spatially. $\mathbf{Y}_1$ is first coded according to the side information from the hyperprior. Then, given the reconstruction $\hat{\mathbf{Y}}_1$, $\mathbf{Y}_2$ is estimated more effectively. Our dual spatial prior shares a similar idea of auto-regressive prior (Minnen et al., 2018)

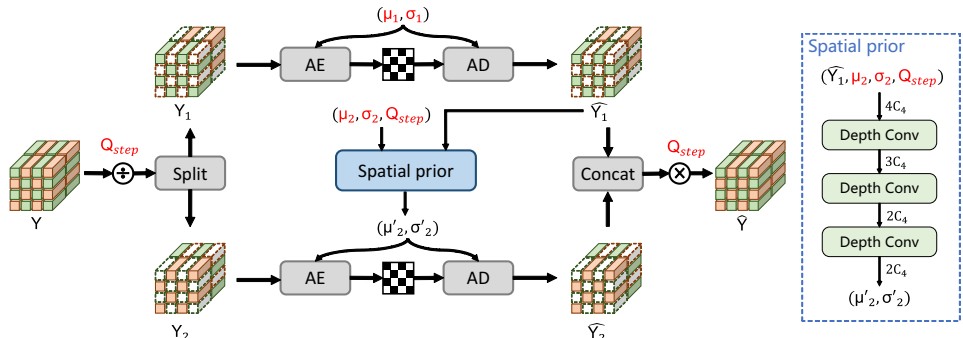

Figure 11: The structure of our dual spatial prior.

that predicts the target conditioning on its neighbor. But ours enjoys parallel computation and a fast inference speed.

**Schemes of models with different capacities.** Tab. 3 summarizes three models used in this paper. We also compare the model size and MACs for $1920 \times 1088$ inputs.

Table 3: Architecture schemes with statistics of models' #Params (MB) and MACs (G). Note that Others include of the hyperprior model, the spatial prior model, and the entropy model. The resolution of the input image is $1920 \times 1088$.

| Model | $C_1, C_2, C_3, C_4$ | | Encoder | Decoder | Others | Total |
|---|---|---|---|---|---|---|
| **L**arge (**L**) | 192, 192, 192, 192 | #Params | 3.19 | 3.38 | 10.82 | 17.38 |
| | | MACs | 549.92 | 538.83 | 76.64 | 1165.39 |
| **M**edium (**M**) | 128, 128, 192, 192 | #Params | 2.08 | 2.33 | 10.82 | 15.23 |
| | | MACs | 262.90 | 258.56 | 76.64 | 598.10 |
| **S**mall (**S**) | 64, 64, 128, 192 | #Params | 0.82 | 1.14 | 10.82 | 12.78 |
| | | MACs | 73.17 | 73.82 | 76.64 | 223.63 |

# B    MASK DECAY

**The pipeline of our model acceleration method.** Figure 12 shows how to accelerate a cumbersome network by our mask decay. First, this cumbersome network is trained and treated as the pretrained teacher. Then, mask layers are inserted in the teacher and are optimized to become sparse. After training, masks are merged into neighbor layers, so that the cumbersome teacher transforms into an efficient student network.

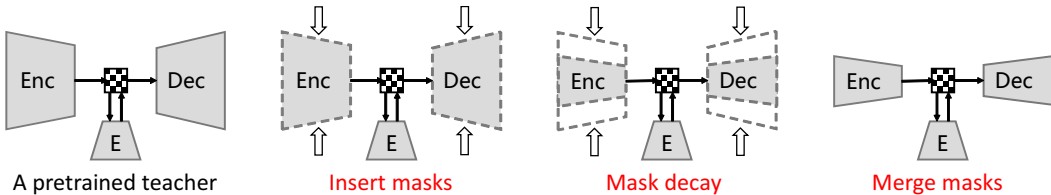

Figure 12: Illustration of our model acceleration process. "Enc" and "Dec" denote the encoder and the decoder, respectively. And "E" represents the entropy model with the hyperprior model.

**Compress one layer's representation by mask decay.** In Fig. 4a, our mask contains $N_2$ parameters which are initialized by 1. After training, $N_s$ parameters remain to be positive while $N_2 - N_s$ parameters become zeros. Then, we conduct two steps to "merge" this mask. First, for $N_2 - N_s$

zero channels, corresponding filters in Conv #1 and #2 are pruned safely. Second, for $N_s$ positive parameters, they are multiplied into Conv #2. After these two steps, the mask can be removed. Claim 1 guarantees that these two steps will not change the outputs. Hence, our sparse mask is able to be eliminated safely.

**Claim 1.** *Our mask can be merged into neighbor convolution layers while keeping the outputs unchanged.*

*Proof.* As showed in Fig. 4a, assume $\boldsymbol{W}_1$ and $\boldsymbol{b}_1$ are parameters in pruning Conv #1, and $\boldsymbol{W}_2$ and $\boldsymbol{b}_2$ are parameters in pruning Conv #2. The LeakyReLU applies the element-wise function:

$$f(x) = \begin{cases} x, & \text{if } x \geq 0, \\ \text{negative\_slope} \times x, & \text{otherwise}. \end{cases} \tag{5}$$

Then given the input $\mathbf{X} \in R^{N_1 \times H \times W}$, the forward computation of the pruning model (the left part in Fig. 4a) is

$$\mathbf{Y}_1 = \boldsymbol{W}_1 \circledast \mathbf{X} + \boldsymbol{b}_1, \qquad (\mathbf{Y}_1 \in R^{N_2 \times H \times W}, \boldsymbol{W}_1 \in R^{N_2 \times N_1 \times 3 \times 3}, \boldsymbol{b}_1 \in R^{N_2}), \tag{6}$$

$$\mathbf{Y}_2 = \mathbf{Y}_1 \odot_c \boldsymbol{m}, \qquad (\mathbf{Y}_2 \in R^{N_2 \times H \times W}, \boldsymbol{m} \in R^{N_2}), \tag{7}$$

$$\mathbf{Y}_3 = f(\mathbf{Y}_2), \qquad (\mathbf{Y}_3 \in R^{N_2 \times H \times W}), \tag{8}$$

$$\mathbf{Y}_4 = \boldsymbol{W}_2 \circledast \mathbf{Y}_3 + \boldsymbol{b}_2, \qquad (\mathbf{Y}_4 \in R^{N_3 \times H \times W}, \boldsymbol{W}_2 \in R^{N_3 \times N_2 \times 3 \times 3}, \boldsymbol{b}_2 \in R^{N_3}), \tag{9}$$

where $\circledast$ denotes the convolution operator, and $\odot_c$ means the channel-wise product. Note that $N_s$ elements of $\boldsymbol{m}$ are positive, while other $N_2 - N_s$ entries are zeros. In the first step, these $N_2 - N_s$ zero channels are pruned safely, and the forward computation becomes

$$\mathbf{Y}_1' = \boldsymbol{W}_1' \circledast \mathbf{X} + \boldsymbol{b}_1', \qquad (\mathbf{Y}_1' \in R^{N_s \times H \times W}, \boldsymbol{W}_1' \in R^{N_s \times N_1 \times 3 \times 3}, \boldsymbol{b}_1' \in R^{N_s}), \tag{10}$$

$$\mathbf{Y}_2' = \mathbf{Y}_1' \odot_c \boldsymbol{m}', \qquad (\mathbf{Y}_2' \in R^{N_s \times H \times W}, \boldsymbol{m}' \in R^{N_s}), \tag{11}$$

$$\mathbf{Y}_3' = f(\mathbf{Y}_2'), \qquad (\mathbf{Y}_3' \in R^{N_s \times H \times W}), \tag{12}$$

$$\mathbf{Y}_4 = \boldsymbol{W}_2' \circledast \mathbf{Y}_3' + \boldsymbol{b}_2 \qquad (\mathbf{Y}_4 \in R^{N_3 \times H \times W}, \boldsymbol{W}_2' \in R^{N_3 \times N_s \times 3 \times 3}, \boldsymbol{b}_2 \in R^{N_3}). \tag{13}$$

Then, in the second step, $N_s$ positive entries in $\boldsymbol{m}'$ will be multiplied into $\boldsymbol{W}_2'$. We have

$$\mathbf{Y}_4 = \boldsymbol{W}_2' \circledast f(\mathbf{Y}_1' \odot_c \boldsymbol{m}') + \boldsymbol{b}_2, \tag{14}$$

$$= \boldsymbol{W}_2' \circledast (f(\mathbf{Y}_1') \odot_c \boldsymbol{m}') + \boldsymbol{b}_2, \tag{15}$$

$$= (\boldsymbol{W}_2' \odot_c \boldsymbol{m}') \circledast f(\mathbf{Y}_1') + \boldsymbol{b}_2. \tag{16}$$

The last equation holds for the linearity property of the convolution operator. Notice that Eq. 16 denotes the computation process in the pruned model (the right part in Fig. 4a). Hence, the pruned model also outputs $\mathbf{Y}_4$ with the input $\mathbf{X}$. That means the outputs are unchanged. By merging the mask $\boldsymbol{m}$, parameters within Conv #1 and # 2 become $(\boldsymbol{W}_1', \boldsymbol{b}_1')$ and $(\boldsymbol{W}_2' \odot_c \boldsymbol{m}', \boldsymbol{b}_2)$, respectively. $\square$

**Mask decay for our encoder and decoder.** How to prune the channels outside the residual connection is also a crucial problem. Fig. 13 shows two manners. Note that Mask #1, #2, and #3 share weights. If we insert Mask #2 and #3 after the residual adding (as in Fig. 13b), the identity output will be degraded by these masks, because most masks are in $[0, 1]$. Inserting masks as in Fig. 13a mitigates this problem and guarantees the identity gradient back-propagation.

## C  THE SCALABLE ENCODER

In this part, we introduce more details for different methods to achieve a scalable encoder.

**End-to-end.** CBANet (Guo et al., 2021) proposed a multi-branch structure for a scalable network. The computation complexity is dynamic by using different numbers of branches in testing. This method can be considered as the baseline in our paper. That is directly training our framework end-to-end (cf. Fig. 14). During training, all encoders generate bitstreams for decoding, and the corresponding loss is calculated. However, this baseline requires a large GPU memory for training and

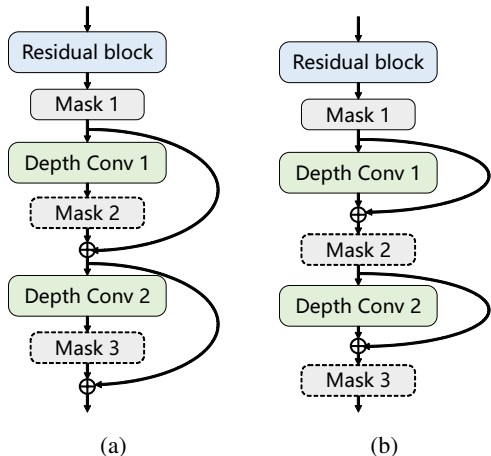

Figure 13: (a) and (b) present two manners to insert masks for the block-level dimensionality. (a) is better because it guarantees the identity gradient back-propagation.

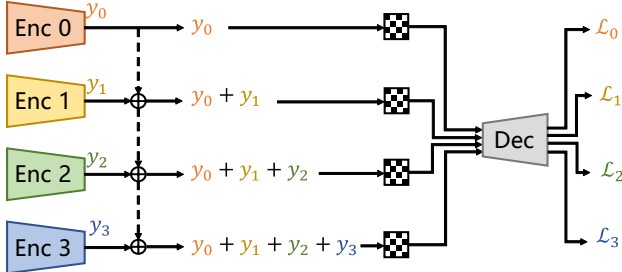

Figure 14: A naive baseline for the scalable encoder trained in an end-to-end manner. Dash lines means detaching gradients.

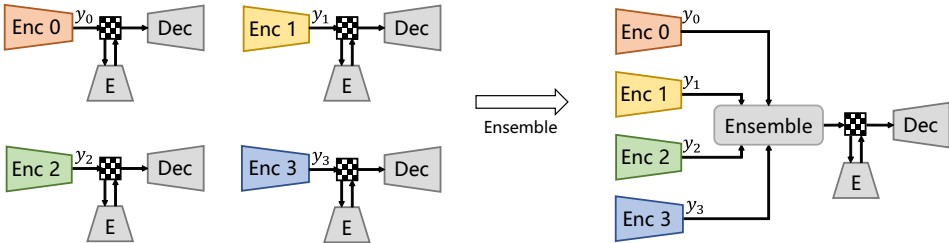

Figure 15: Illustration of training separate small encoders to achieve a scalable encoder.

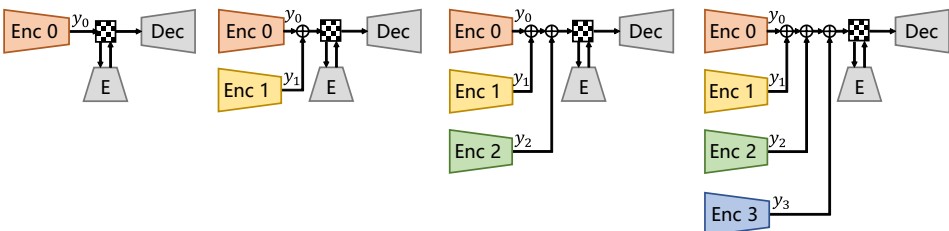

Figure 16: Illustration of training small encoders one-by-one to achieve a scalable encoder. In each time, the new small encoder is encouraged to learn residual representations.

Table 4: BD-Rate (%) comparison for PSNR on Kodak. The encoder is scalable and the decoder is Large. MACs in the table are for encoding with $1920 \times 1088$ resolution, while MACs for other modules is 615.47 G in total. The anchor is VTM.

|  | $S_0$ | $S_{0,1}$ | $S_{0,1,2}$ | $S_{0,1,2,3}$ | L |
|---|---|---|---|---|---|
| MACs (G) | 73.17 | 146.34 | 219.51 | 292.68 | 549.92 |
| End-to-end | 3.1 | 2.2 | 1.8 | 1.6 | |
| Separate | 2.8 | 1.9 | 1.7 | 1.5 | -1.1 |
| One-by-one | 2.8 | 1.4 | 0.9 | 0.6 | |
| Ours | **1.4** | **0.4** | **-0.1** | **-0.2** | |

performs badly demonstrated by our experiments. End-to-end training requires optimizing all parameters of the whole network. And it does not utilize the knowledge in the pretrained cumbersome model. Hence, the training is inefficient.

**Separate.** Another baseline method is directly training separate small encoders with a uniform decoder. As shown in Fig. 15, four small encoders are trained with the frozen decoder and entropy model. For testing, different numbers of encoders can be chosen for the specific encoding complexity. In the right part of Fig. 15, all four encoders are chosen. The ensemble module will receive all bitstreams from encoders and choose the best bitstream to send. Note that the raw image is available in encoding. The ensemble module is able to reconstruct each bitstream and compute the RD score according to the raw image. However, these separate encoders suffer from the homogenization problem. With the uniform decoder, these encoders generate similar bitstreams. In Fig. 1b, we find that adding more encoders improves the RD performance marginally.

**One-by-one (RRL).** We propose residual representation learning (RRL) to mitigate the homogenization problem. As illustrated in Fig. 16, we train small encoders progressively. In each step, a new small encoder is added and trained from scratch. Note that other parts are frozen. The new encoder is encouraged to learn the residual representations which effectively alleviates the homogenization problem and enlarges the diversity of these small encoders. Fig. 1b shows "one-by-one" outperforms "separate encoders" by a significant margin.

**Ours (RRL + mask decay).** At last, mask decay further improves the RD performance. As illustrated in Fig. 5, mask decay directly utilizes knowledge of the cumbersome model. In each step, the small encoder is obtained by pruning from the cumbersome teacher. Here the teacher means the large encoder in Tab. 3, whereas the student means the small encoder. In Fig. 5, the encoder with dash boundaries denotes the teacher encoder equipped with masks. During training, these masks will become sparse so that the network is able to transform into the student structure (i.e., the student encoder). Both RRL and mask decay treat the cumbersome teacher as a reference, which makes the training more effective. Fig. 1b shows "ours" (RRL + mask decay) achieves superior RD performance and significantly narrows the performance gap from the teacher.

Tab. 4 summarizes results for our scalable encoders with the large decoder. We also conduct experiments for the small decoder, and Tab. 5 summarizes results. In these tables, $S_{0,...,i}$ denotes using $i + 1$ small encoders, while L means the raw large encoder.

Table 5: BD-Rate (%) comparison for PSNR on Kodak. The encoder is scalable and the decoder is Small. MACs in the table are for encoding with $1920 \times 1088$ resolution, while MACs for other modules is 150.46 G in total. The anchor is VTM.

|  | $S_0$ | $S_{0,1}$ | $S_{0,1,2}$ | $S_{0,1,2,3}$ | L |
|---|---|---|---|---|---|
| MACs (G) | 73.17 | 146.34 | 219.51 | 292.68 | 549.92 |
| End-to-end | 9.1 | 8.6 | 8.2 | 7.9 | |
| One-by-one | 7.3 | 6.3 | 5.8 | 5.5 | 4.0 |
| Ours | **6.3** | **5.6** | **5.0** | **4.8** | |

## D    DETAILS FOR EXPERIMENTS

**Training.** The training dataset is the training part of Vimeo-90K (Xue et al., 2019) septuplet dataset. During training, the raw image is randomly cropped into $256 \times 256$ patches. For a fair comparison, all models are trained with 200 epochs. For our method, it cost 60 epochs for the teacher to transform into the student with mask decay, then the student is finetuned by 140 epochs. AdamW is used as the optimizer with batch size 16. The initial learning rate is $2e-4$, and decays by 0.5 at 50, 90, 130, 170 epochs. The default decay rate $\eta$ for our mask decay is $4e-5$. All models are trained on a computer with 8 V100 GPUs. Our large and small model cost about 33 and 16 hours for training, respectively.

**Testing.** BD-Rate (Bjontegaard, 2001) for peak signal-to-noise ratio (PSNR) is our main metric to compare different models. We use VTM 17.2 as the anchor, while negative numbers indicate bitrate saving and positive numbers indicate bitrate increase. Testing datasets contain Kodak (Franzen, 1999), HEVC test sequences (Sharman & Suehring, 2017), Tecnick (Asuni & Giachetti, 2014). Kodak contains 24 images with the $768 \times 512$ resolution. And Tecnick consists of 100 images with the $1200 \times 1200$ resolution. HEVC is a video dataset that consists of 4 classes: B (1080P), C (480P), D (240P), E (720P). For testing image codecs on HEVC, only the first frame of each video sequence is used. We report BD-Rate on each HEVC class and average all for convenient comparison. For one testing image, we pad zeros on the boundary to make the resolution as multiples of 64. We test the latency on two computers. One is equipped with one 2080Ti GPU and two Intel(R) Xeon(R) E5-2630 v3 CPUs. Another is equipped with one A100 GPU and one AMD Epyc 7V13 CPU. Both encoding and decoding times are gathered. Note that the latency *contains arithmetic coding*, but ignores the disk I/O time.

## E    IMPLEMENT DETAILS FOR OTHER IMAGE CODECS

**VTM-17.2**[1] is the reference software for the state-of-the-art traditional codec (H.266/VVC), and it is released on Jul 17, 2022. We use CompressAI-1.1.0[2] to gather evaluation results. We install these two softwares according to the official instructions. The default configuration of VTM is used. All results are gathered by running the following command:

```
python –m compressai.utils.bench vtm [path to image folder] –c [path to  VTM folder]/cfg/
    encoder_intra_vtm.cfg –b [path to  VTM folder]/bin –j  8 –q 24 26 28 30 32 34 36 38
    40 42
```

**Entroformer**[3] is proposed in Qian et al. (2022). Two series models are released: parallel, non-parallel. We evaluate the RD performance of the parallel model on Tecnick and Kodak, respectively. But when we measure its latency, we find it requires a large GPU memory for generating bitstreams.

---

[1]https://vcgit.hhi.fraunhofer.de/jvet/VVCSoftware_VTM/-/releases/VTM-17.2

[2]https://github.com/InterDigitalInc/CompressAI/releases/tag/v1.1.0

[3]https://github.com/damo-cv/entroformer

Table 6: Latency (ms) comparison with SwinT. Lower means better. To compare with SwinT, we test ours on 2080Ti with $768 \times 512$ inputs, and directly cite their numbers. Note that SwinT does not reconstruct images for encoding, whereas ours reconstructs images.

| Latency | SwinT | | | | EVC | | |
|---|---|---|---|---|---|---|---|
| | SwinT-ChARM | SwinT-Hyperprior | Conv-ChARM | Conv-Hyperprior | Large | Medium | Small |
| encoding | 135.3 | 100.5 | 95.7 | 62.0 | 63.0 | 44.7 | 28.4 |
| decoding | 167.3 | 114.0 | 264.0 | 219.3 | 41.1 | 32.4 | 24.4 |

**STF**[4] is proposed in Zou et al. (2022). The authors release two architectures: CNN-based model and Transformer-based model. But, only two pretrained models for two qualities are available. We evaluate one model on Tecnick and cite numbers in their paper for Kodak.

**SwinT-ChARM** is proposed in Zhu et al. (2022). Because the authors did not release their code, we cite numbers in their paper and the webpage[5]. The latency is reported in their paper, and Tab. 6 compares ours with theirs. Our models achieve lower latency.

**ELIC** is proposed in He et al. (2022). Uneven channel-conditional adaptive coding is proposed in this work. He et al. (2022) designed two models ELIC and ELIC-sm, where ELIC-sm is adapted from ELIC with a slim architecture. Because the authors did not release their code, we make comparisons with numbers in their paper.

**Cheng2020** is proposed in Cheng et al. (2020). Attention modules and Gaussian Mixture Model are proposed to improve the RD performance. Due to its high impact, it is always considered as the baseline method for neural image codecs. However, the autoregressive module is adopted in their raw model. And inference on GPU is not recommended for the autoregressive models. When testing on CPU, it takes more than one second to process one image. He et al. (2021) proposed the checkerboard module to mitigate this problem. We compare ours with this model in Tab. 7.

**Xie2021** is proposed in Xie et al. (2021). The invertible neural networks are proposed and used as the stronger backbone. But the same as Cheng2020, Xie2021 is also equipped with the autoregressive module. It suffers from high latency due to this module. We compare ours with this model in Tab. 7.

**Minnen2018 and Minnen2020** are proposed in Minnen et al. (2018) and Minnen & Singh (2020), respectively. The autoregressive context model was first introduced in Minnen et al. (2018) to boost the RD performance of neural image codecs. Minnen & Singh (2020) further improved it by considering channel-wise redundancy. Both two works have a significant impact in the neural image compression and are always considered as the baseline method for comparisons. We also compare ours with them in Tab. 7.

**Ballé2018** is proposed in Ballé et al. (2018). The hyperprior structure was introduced, which is a milestone in the field of image codec. Most latter models adopt this structure, including ours. Without a context module, it achieves a high inference speed. But without using recent proposed techniques, it suffers from a poor RD performance compared with recent SOTA models.

## F MORE EXPERIMENTAL RESULTS

**Comparison to more methods.** He et al. (2022) proposed two models for efficient learned image compression, namely ELIC and ELIC-sm, respectively. However, they did not release their codes and models and it is difficult for us to make a fair comparison. We follow their testing manner to compare numbers in their paper. First, in their supplementary material, they mentioned the deterministic inference mode was not enabled when testing the model speeds. We test latencies in deterministic/non-deterministic modes, respectively. We find that non-determinism reduces the latency by average 16% but easily results in catastrophic failures for decoding (Ballé et al., 2019). Second, different from testing end-to-end latency with arithmetic coding in our main paper, here we test each module's latency as that in He et al. (2022). Hence, without considering arithmetic coding,

---

[4]`https://github.com/Googolxx/STF`
[5]`https://openreview.net/forum?id=IDwN6xjHnK8`

Table 7: BD-Rate (%) and inference latency (ms) comparisons. BD-Rate is computed from PSNR-BPP curve of Kodak over the anchor VTM. Lower BD-Rate means better. Our models were tested in deterministic/non-deterministic modes, respectively. Other methods were tested by He et al. (2022) with the non-deterministic mode.

| GPU | Model | BD-Rate | Inference Latency | | | | | |
| --- | --- | --- | --- | --- | --- | --- | --- | --- |
| | | | Tot. | YEnc. | YDec. | ZEnc. | ZDec. | Param. |
| TITAN Xp | Ballé2018 (Ballé et al., 2018) | 40.85 | 30.3 | 11.9 | 16.1 | 1.2 | 1.1 | - |
| | Minnen2018 (He et al., 2021) | 20.00 | 37.6 | 11.6 | 17.3 | 1.5 | 1.5 | 5.7 |
| | Cheng2020 (He et al., 2021) | 3.89 | 90.8 | 36.8 | 44.2 | 0.9 | 1.2 | 7.7 |
| | Minnen2020 (Minnen & Singh, 2020) | 1.11 | 77.7 | 10.6 | 15.4 | 1.4 | 1.5 | 48.8 |
| | Xie2021 (Xie et al., 2021) | -0.54 | $> 10^3$ | 58.0 | 162.2 | 1.3 | 1.3 | $> 10^3$ |
| | ELIC-sm (He et al., 2022) | -1.07 | 41.5 | 12.6 | 17.4 | 1.1 | 1.4 | 9.0 |
| | ELIC (He et al., 2022) | -7.88 | 82.4 | 31.2 | 38.3 | 1.2 | 1.6 | 10.1 |
| 2080Ti | EVC-SS (deterministic) | 6.9 | 22.1 | 4.2 | 7.4 | 1.4 | 5.4 | 3.7 |
| | EVC-SS (non-deterministic) | 6.9 | 17.9 | 3.8 | 5.7 | 0.9 | 3.7 | 3.8 |
| | EVC-MM (deterministic) | 0.9 | 31.4 | 9.1 | 11.9 | 1.4 | 5.3 | 3.7 |
| | EVC-MM (non-deterministic) | 0.9 | 26.5 | 8.3 | 10.2 | 0.9 | 3.5 | 3.6 |
| | EVC-LL (deterministic) | -1.1 | 44.5 | 15.5 | 18.6 | 1.4 | 5.3 | 3.7 |
| | EVC-LL (non-deterministic) | -1.1 | 38.4 | 14.1 | 16.2 | 0.9 | 3.6 | 3.6 |

Table 8: Speed (megapixel/s) comparison. Higher means better. Entroformer (Qian et al., 2022) and STF (Zou et al., 2022) are the SOTA methods. 'OM' means out of memory. Note that 2080Ti and A100 have 12GB and 80GB memory, respectively.

| Resolution | GPU | Type | Entroformer | STF | | EVC | | |
| --- | --- | --- | --- | --- | --- | --- | --- | --- |
| | | | | Transformer | CNN | Large | Medium | Small |
| $768 \times 512$ | 2080Ti | encoding | OM | 2.23 | 2.48 | 6.24 | 8.80 | 13.85 |
| | | decoding | OM | 1.94 | 1.87 | 9.57 | 12.14 | 16.12 |
| | A100 | encoding | 0.48 | 3.39 | 4.08 | 18.64 | 19.86 | 22.22 |
| | | decoding | 0.09 | 2.75 | 3.33 | 20.59 | 23.00 | 25.21 |
| $1920 \times 1080$ | 2080Ti | encoding | OM | 3.60 | 4.55 | 6.79 | 11.42 | 22.81 |
| | | decoding | OM | 3.90 | 3.18 | 11.57 | 17.56 | 28.33 |
| | A100 | encoding | 0.27 | 5.83 | 7.46 | 24.63 | 36.83 | 66.04 |
| | | decoding | OM | 5.84 | 7.36 | 34.45 | 44.59 | 69.82 |

the latency is lower than that in our main paper. Third, unfortunately, we do not have TITAN Xp[6] GPUs, and we test our models on 2080Ti[7].

Tab. 7 summarizes results. For our models, "S", "M", and "L" denote small, medium, large, respectively. We use two letters to represent the encoder and the decoder, respectively. Compared with previous SOTA methods (Ballé2018, Minnen2018, Cheng2020, Minnen2020, Xie2021), our large model ("EVC-LL") achieves better RD performance and inference latency. And it is also comparable to ELIC-sm. Note that ours enjoys a single model to handle all RD trade-offs, while ELIC trained different models for each trade-off.

**Speed (megapixel/s) comparison.** In Tab. 8, we also compute models' speeds as megapixel/s according to Tab. 1. Our models outperform previous models for this metric, too.

**Comparison for our different models.** Fig. 17 presents RD curves on Kodak. Our mask decay "MD" improves our baseline models significantly. In these figures, "L", "M", and "S" denote Large,

---

[6]https://www.nvidia.com/en-us/titan/titan-xp/
[7]https://www.nvidia.com/en-us/geforce/graphics-cards/compare/?section=compare-20

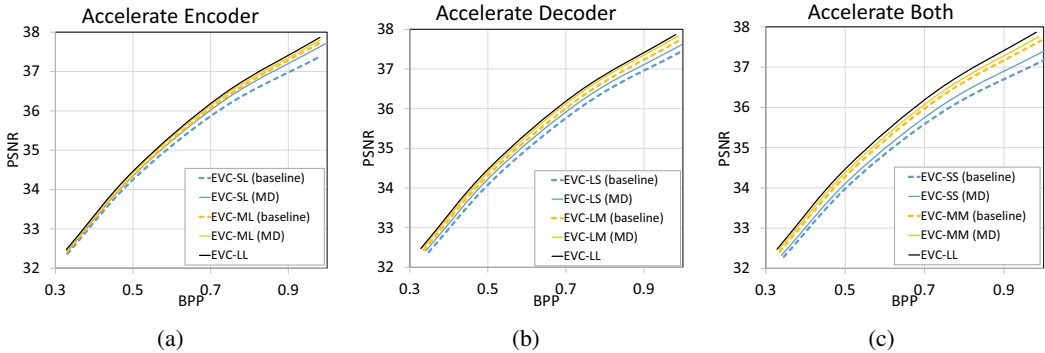

Figure 17: RD curves on Kodak for our models with different complexities. "MD" denotes mask decay. (a) and (b) show curves for accelerating only encoder or decoder, respectively. (c) presents curves for accelerating both encoder and decoder.

Table 9: BD-Rate (%) comparison for PSNR on CLIC2021 and Tecnick. L, M, and S denote Large, Medium, and Small, respectively. The anchor is VTM. Lower BD-Rate means better. MACs is for $1920 \times 1088$ inputs.

| Enc | Dec | MACs (G) | Method | CLIC2021 | Tecnick |
|-----|-----|----------|--------|----------|---------|
| L | L | 1165.39 | Baseline | -0.7 | -2.0 |
| M | L | 878.37 | Baseline | 1.0 | -0.0 |
|   |   |  | Mask Decay | **0.6 (24%↑)** | **-0.3 (15%↑)** |
| S | L | 688.64 | Baseline | 4.3 | 1.9 |
|   |   |  | Mask Decay | **1.9 (48%↑)** | **0.2 (44%↑)** |
| L | M | 885.12 | Baseline | 2.4 | 1.0 |
|   |   |  | Mask Decay | **0.9 (48%↑)** | **-0.3 (43%↑)** |
| L | S | 700.38 | Baseline | 7.1 | 5.4 |
|   |   |  | Mask Decay | **4.3 (36%↑)** | **2.7 (36%↑)** |
| M | M | 598.10 | Baseline | 4.6 | 2.6 |
|   |   |  | Mask Decay | **1.6 (57%↑)** | **0.7 (41%↑)** |
| S | S | 223.63 | Baseline | 11.6 | 9.9 |
|   |   |  | Mask Decay | **8.5 (25%↑)** | **7.6 (19%↑)** |

Medium, and Small, respectively. And "EVC-SL" means the encoder is small and the decoder is large.

Tab. 9 presents these model's results on CLIC2021[8] and Tecnick. Mask decay outperforms the baselines by significant margins.

**Sparsity losses and decay rates.** For easier to read, we replicate Fig. 7 for larger sizes. Figs. 18 and 19 are these replications.

**Disscusions for important channels.** Tab. 10 presents the specific numbers in Fig. 8a. There are four masks in the first two blocks. We analyze the first mask (ResB.1.mask1) for example. Mask decay is required to choose 64 channels from the teacher's 192 channels. With four different decay rates, only 83 channels are considered and they choose 44 channels together. It seems like these channels are important for superior performance. We treat these 83 channels as $\mathbb{C}_F$ and let our algorithm avoid choosing these channels. A1B (avoid specific channels in the first block) only selects 2 channels in $\mathbb{C}_F$, while A2B selects 3. First, it demonstrates our AMD indeed avoids specific channels. Next, we compare the RD performance of these methods. Surprisingly, AMD

---

[8]https://storage.googleapis.com/clic2021_public/professional_test_2021.zip

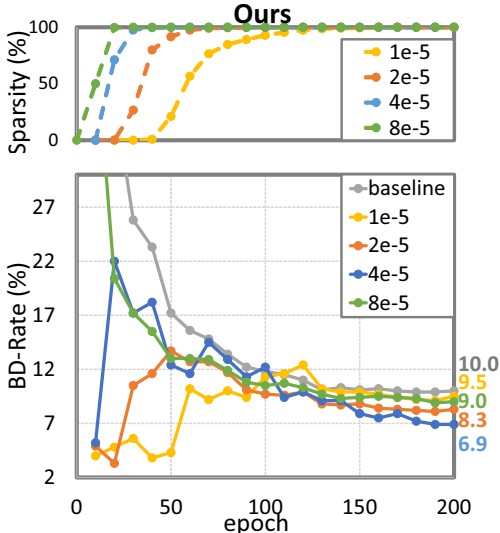

Figure 18: Mask decay with our sparsity loss (Eq. 2). Different decay rates $\eta$ were tried. This figure is the replication of Fig. 7a.

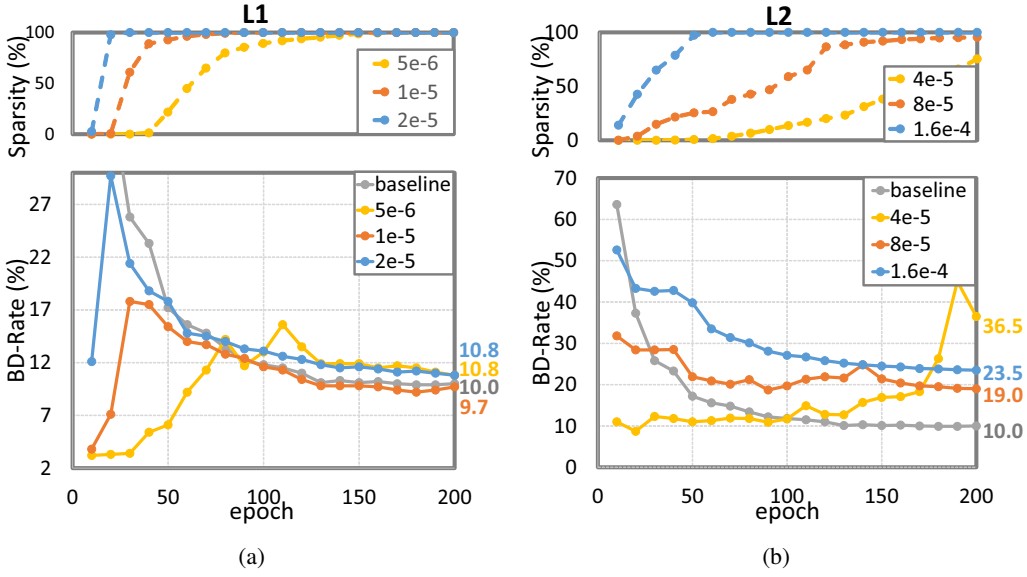

Figure 19: Mask decay with L1 and L2 losses. We adjusted the decay rate $\eta$ for each criterion. These two figures are replications of Fig. 19a and 19b.

Table 10: The relationship for channels chosen by different settings. "ResB.1" and "DCB.2" means the Residual block and the Depth Conv block, respectively. Note that they are the first two blocks in our model. $\mathbb{C}$ denotes the set of chosen channels, and $F = \{1e-5, 2e-5, 4e-5, 8e-5\}$ which indicates different decay rates in Figure 7a. A1B and A2B means avoiding specific channels in the first 1 and 2 blocks, respectively. We report the cardinality of each set.

| Mask Layer | Teacher | Student | $\cap_{i \in F} \mathbb{C}_i$ | $\cup_{i \in F} \mathbb{C}_i$ | $\mathbb{C}_{A1B} \cap \mathbb{C}_F$ | $\mathbb{C}_{A1B} \cup \mathbb{C}_F$ | $\mathbb{C}_{A2B} \cap \mathbb{C}_F$ | $\mathbb{C}_{A2B} \cup \mathbb{C}_F$ |
|---|---|---|---|---|---|---|---|---|
| ResB.1.mask1 | 192 | 64 | 44 | 83 | 2 | 145 | 3 | 144 |
| ResB.1.mask2 | 192 | 64 | 47 | 83 | 0 | 147 | 2 | 145 |
| DCB.2.mask1 | 192 | 64 | 44 | 86 | 56 | 94 | 7 | 143 |
| DCB.2.mask3 | 768 | 256 | 179 | 328 | 191 | 393 | 12 | 572 |

results in comparable performance. That indicates that there are no important channels for superior performance.

## G    VISUALIZATIONS

Fig. 20 visualizes our models' reconstructions. Comparing our small model (EVC-SS) and large model (EVC-LL), the reconstructions are almost the same. That demonstrates our mask decay does not result in extra artifacts. Compared with VTM, ours enjoy fast encoding and decoding.

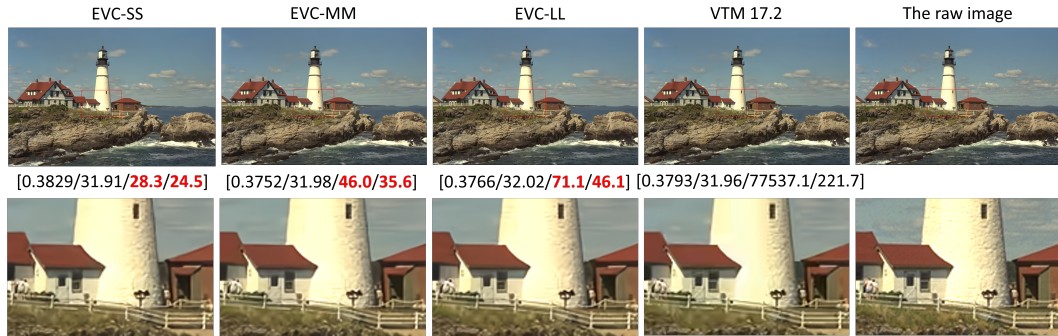

Figure 20: Visualization of our models' reconstruction. EVC-SS denotes our model equipped with the small encoder and the small decoder, while M and L means medium and large, respectively. Numbers in the tuple are BPP, PSNR, the encoding time (ms), and the decoding time (ms), respectively. Note that the latency is measured on a computer with 2080Ti GPU. Our models are dramatically faster than VTM.

