# OpenReview forum: "EVC: Towards Real-Time Neural Image Compression with Mask Decay"
_ICLR.cc/2023/Conference — ICLR 2023 poster_

### Official Review · Reviewer_RU83 · 2022-10-23

**Confidence:** 4
**Correctness:** 4
**Technical Novelty And Significance:** 3
**Empirical Novelty And Significance:** 2
**Recommendation:** 6

**Clarity, Quality, Novelty And Reproducibility:**

Overall, the paper is clearly written with high quality in terms of the presentation and evaluation. I don't have major concerns about reproducibility, especially since the authors say code will be released (mentioned at the end of the abstract).

I do think clarity could be improved in terms of the scalable encoder. Although the technical approach seems clear from the description and Fig 5, it's not clear to me how it relates to the small/medium/large architectures (discussed at the end of Section 3.1) and mask decay. Are these different methods -- that is, you *either* use RRL or mask decay? Or is mask decay applied on top of RRL as the arrows seem to imply in Fig 1b? If the latter, more detail on how mask decay (a version of student-teacher learning) is applied is needed, e.g. it would seem that the teacher (the "cumbersome" model) would need to learn N residual encoders and then each of those would be masked to yield a smaller student.

Also, I think that where the authors say the method "achieves 30 FPS for the 1920 x 1080 inputs", the results should be given as megapixel/s (mp/s) instead. This is an image compression model so there are no FPS per se, and mp/s (or mb/s though I assume all results here use 8 bits per channel) is the standard unit.

**Strength And Weaknesses:**

The paper is well-motivated, and the claims are empirically evaluated and compared against strong/appropriate baselines.

The mask decay idea, which combines a novel sparsity loss applied via weight decay, addresses a significant problem for neural image compression. Other papers have looked at methods for speeding up neural codecs by reducing the channel depth of conv layers (the primary runtime bottleneck), but the RD penalty is typically quite high. The authors show that mask decay has a much smaller penalty (Fig 1b) and thus provides a new approach to close the runtime gap between learning-based and hand-engineered codecs.

The main weakness of the paper is empirical. The authors should compare against this paper from CVPR 2022:

ELIC: Efficient Learned Image Compression with Unevenly Grouped Space-Channel Contextual Adaptive Coding
Dailan He, Ziming Yang, Weikun Peng, Rui Ma, Hongwei Qin, Yan Wang
https://arxiv.org/abs/2203.10886

This paper explores a combination of channel-wise and spatial decomposition for the entropy model that's different than what's used in the paper under review (primarily, it uses five channel-wise groups with uneven depths). The authors show that this allows them to reduce the computational complexity of the analysis and synthesis transforms and achieve a net benefit in terms of RD performance vs. runtime. In particular, I think the results is both faster and gives a slightly better compression rate than what's presented here (it's hard to be 100% sure without overlaying the RD graphs, and runtime comparisons are difficult since different hardware was used (2080Ti vs Titan Xp)).

Regardless, I think there are enough novel ideas with promising impact on the neural image compression subfield that this paper should still be accepted.

**Summary Of The Paper:**

This paper explores neural image compression and focuses on improving existing methods in several ways:

1. A single model that allows variable bit rate encodings, variable encode compute, and faster run time. Although previous research has explored these items in isolation, I do not know of a model that incorporates all of them.

2. Better model pruning through a novel sparsity loss ("mask decay"). This loss (applied via weight decay) leads to better rate-distortion (RD) loss than training models with identical architectures from scratch or by using simpler sparsity losses like L1 and L2.

3. Scalable encoding, in terms of computation, is achieved through residual representation learning (RRL), where the encoder is iteratively refined by learning a mapping from the input image to a residual in latent space (Fig 5). At inference time, the encoder can decide how many of the residual encoders to run to trade off runtime for RD performance.



**Summary Of The Review:**

I'm recommending that this paper be accepted based on promising results from the novel sparsity loss and mask decay. This approach can be applied to other models and appears to give much better results than previous pruning methods applied to compression models.

As mentioned above, a comparison to ELIC should be added, and a favorable comparison (runtime and RD) would strengthen my recommendation based on the empirical results.

---

> ### Author Response · Authors · 2022-11-11
> **To Reviewer RU83**
>
> Thanks for your review and appreciation!
>
> > ``Comparison to ELIC''
>
> We also appreciate ELIC [1] which is a powerful and efficient neural image compression model. Actually, we have tried to reimplement their method before our submission. However, we have not achieved the performance reported in their paper, mainly due to our problems. Hence, it is regrettable that we cannot make a fair comparison with them in our paper. As suggested by reviewers, we would like to follow their settings and compare ours with the numbers in their paper. First, in their supplementary material, they mentioned the deterministic inference mode was not enabled when testing the model speeds. We test latencies in deterministic/non-deterministic modes, respectively. We find that non-determinism reduces the latency by average 16% but easily results in catastrophic failures for decoding [2]. Second, different from testing end-to-end latency with arithmetic coding in our main paper, here we test each module's latency, same as that in [1]. Hence, without considering arithmetic coding, the latency is lower than that in our main paper. Third, unfortunately, we do not have TITAN Xp GPUs, and we test our models on 2080Ti.
>
> We summarize the results in Table 7. Our large model (-1.1% BD-Rate with 38.4ms inference latency) is comparable to ELIC-sm (-1.07% BD-Rate with 41.5ms inference latency) for the RD performance and inference latency. Note that ours enjoys a single model to handle all RD trade-offs, while ELIC trained different models for each trade-off. And we believe single model with variable-bit-rate is more practical in real-world applications.
>
> [1] Dailan He, Ziming Yang, Weikun Peng, Rui Ma, Hongwei Qin, and Yan Wang.   ELIC: Efficient learned image compression with unevenly grouped space-channel contextual adaptive coding. In Proceedings of the IEEE/CVF Conference on Computer Vision and Pattern Recognition, pp.5718–5727, 2022.
>
> [2] Johannes Balle, Nick Johnston, and David Minnen. Integer networks for data compression with latent-variable models. In International Conference on Learning Representations, pp. 1–10, 2019.
>
> > ``Clarify the scalable encoder''
>
> Thanks for this suggestion. We have updated Appendix C by including more details. In Fig.5, the gray encoder with dash boundaries denotes the large encoder equipped with masks, while the color encoder means the small encoder. RRL means training each encoder by learning the residual representations, and it focuses on how to aggregate these encoders. In Fig.1(b), the ``one-by-one'' curve denotes the results of RRL. We directly train these small encoders progressively (cf. Fig.16). Mask decay means training a small encoder (student) from a large encoder (teacher). These two techniques are compatible (as in Fig.5). In short, we train each encoder by mask decay and aggregate all encoders by RRL. Overall, these two techniques treat the teacher as a reference implicitly, therefore making training more effective. And our scalable encoder significantly narrows the performance gap between the teacher.
>
> > ``FPS vs. megapixel/s''
>
> Thanks for this suggestion. We compute models' speeds as megapixel/s and report them in Table 8. Our models also outperform previous models for this metric. We find that a larger resolution always results in a higher megapixel/s. For example, on 2080Ti, the encoding speeds of our small model are 13.85/22.81 megapixel/s for $768\times 512$/$1920\times 1080$, respectively. We guess it is because a large-resolution image will improve the GPU utilization rate.

---

### Official Review · Reviewer_7uFq · 2022-10-24

**Confidence:** 4
**Correctness:** 3
**Technical Novelty And Significance:** 2
**Empirical Novelty And Significance:** 2
**Recommendation:** 3

**Clarity, Quality, Novelty And Reproducibility:**

The novelty of this paper is not significant considering the dynamic complexity that has been investigated by existing research.

**Strength And Weaknesses:**

**Strengths**

1. The proposed method is useful for practical neural image compression. The Mask decay solution seems to be reasonable and interesting.
2. The inference speed reported by this paper is encouraging.  And dynamic complexity is an interesting design in practical applications.

**Weaknesses**

1. Several strong baseline methods[1,2,3] should be discussed and compared. In particular, ELIC[1] method is also proposed for practical image compression and the authors are suggested to make further comparisons.

     [1] He, Dailan, Ziming Yang, Weikun Peng, Rui Ma, Hongwei Qin, and Yan Wang. “ELIC: Efficient Learned Image Compression with Unevenly Grouped Space-Channel Contextual Adaptive Coding.” arXiv, March 29, 2022. http://arxiv.org/abs/2203.10886.

      [2] Xie, Yueqi, Ka Leong Cheng, and Qifeng Chen. “Enhanced Invertible Encoding for Learned Image Compression.” ArXiv:2108.03690 [Cs, Eess], August 8, 2021. http://arxiv.org/abs/2108.03690.

      [3] Kim, Jun-Hyuk, Byeongho Heo, and Jong-Seok Lee. “Joint Global and Local Hierarchical Priors for Learned Image Compression.” In 2022 IEEE/CVF Conference on Computer Vision and Pattern Recognition (CVPR), 5982–91. New Orleans, LA, USA: IEEE, 2022. https://doi.org/10.1109/CVPR52688.2022.00590.

2. Using the transformer based NIC methods as baseline methods is not a good choice. Most transformer based approaches suffer from the high complexity cost.

3. For the proposed mask decay approach, the improvements on the kodak dataset are not significant. It brings 1-3% bitrate savings.

4. The comparison between the proposed method and SlimCAE is not fair since they use different backbone networks.

5. The novelty of dynamic complexity is not significant as several approaches like SlimCAE and CBA-NET[4] have been proposed. The authors should make further explanations.

     [4] Cbanet: Towards complexity and bitrate adaptive deep image compression using a single network.

6. The training strategy seems a little tricky. For example, how to decide the termination of mask decay. The authors are suggested to provide more details.





**Summary Of The Paper:**

This paper proposed a real-time neural image compression solution. This approach utilizes the knowledge distillation approach to obtain low-complexity codecs through the mask decay design. Furthermore, the paper also introduces the scalable encoder for neural image compression and achieves the dynamic complexity for different latency requirements. Experimental results on several benchmark datasets are provided to demonstrate the superiority of the proposed approach. In summary, the paper focuses on practical neural image compression based on knowledge distillation and slimmable design.

**Summary Of The Review:**

The paper focuses on practical neural image compression based on the KD technique and dynamic complexity. The selection of baseline methods is not convincing and several strong baseline methods are missing.

---

> ### Author Response · Authors · 2022-11-11
> **To Reviewer 7uFq**
>
> > ``Novelty''
>
> We want to clarify our novelties and contributions first. They can be summarized as three folds. First, we propose an efficient single-model variable-bit-rate network with a high RD performance. This denotes three important factors in practice: speed, footprint, and RD performance. To the best of our knowledge, most recent works consider at most two aspects, whereas we consider all. Second, for the first time, we study the KD techniques to improve a student image compression model. Note that there is no paper to successfully enable KD on image neural compression models. We also propose a novel sparsity loss to mitigate the drawbacks of Lp losses. Third, we further enable encoding scalability. With residual representation learning and mask decay, our scalable encoder significantly narrows the performance gap between the teacher.
>
> > ``Comparisons between ours and SlimCAE/CBANet''
>
> Thank you for providing the reference for CBANet. CBANet achieves dynamic complexity by a multi-branch structure, while SlimCAE adopted slimmable layers (e.g., slimmable convolution layers). The multi-branch structure can be considered as the baseline method (``End-to-end'') in our paper. Our method outperforms it with a significant margin. That is because our RRL and mask decay make training more effective. Compared with SlimCAE, our small encoders share the same simple architecture, which is easier for engineering optimization in real-world applications compared to the slimmable convolution. Note that they did not release their codes and models. To make a comparison, we asked the authors for RD numbers via emails.
>
> [1] Jinyang Guo, Dong Xu, and Guo Lu. CBANet: Towards complexity and bitrate adaptive deep image compression using a single network. arXiv preprint arXiv:2105.12386, 2021.
>
> > ``Comparisons with Transformer based methods''
>
> In our paper, Entroformer and STF were taken for comparison. The main reason is they are the latest methods with public codes and models. Note that STF has also tried CNNs as the backbone (cf. Tab. 1). Our models are faster than STF (CNN) significantly. For encoding one 768x512 image on 2080Ti, our large model only needs 63.0ms, while STF (CNN) costs 158.5ms.
>
> > ``Compared to ELIC and more baselines''
>
> Thank you for providing these references. All of them have been included in our paper. Xie2021 [1] proposed the invertible neural network that effectively improves the neural image compression model. Informer [2] firstly proposed using an attention mechanism to exploit both global and local information. ELIC [3] proposed uneven channel-conditional adaptive coding to improve the coding performance without damage to running speed. For the **speed**, Xie2021 suffers from high latency due to the autoregressive module, while both Informer and ELIC did not release codes and models. Informer did not report their latency in paper. For ELIC, we have tried to reimplement their method before our submission. However, we have not achieved the performance reported in their paper, mainly due to our problems. Hence, we test our models on our computer following their testing manner. At last, our large model (38.4ms) is comparable to ELIC-sm (41.5ms) for the inference latency.  For the **RD performance**, both Xie2021 and ELIC-sm outperforms VVC by -0.54% and -1.07% BD-Rates, respectively. Informer was mainly compared with BPG in their paper, without reporting results over VVC. Our large model achieves -1.1% BD-Rate over VVC, on par with Xie2021 and ELIC-sm.  For the **footprint**, all these three methods train separate models for each RD trade-off, while ours train one model. And we believe single model with variable-bit-rate is more practical in real-world applications. More experimental details and results are in Appendix F.
>
> Overall, we appreciate ELIC achieves remarkable efficiency and RD performance. Our model is on par with it for these two aspects, and enjoys a single-model variable-bit-rate framework.
>
> [1] Yueqi Xie, Ka Leong Cheng, and Qifeng Chen. Enhanced invertible encoding for learned image compression. In Proceedings of the 29th ACM International Conference on Multimedia, pp. 162–170, 2021.
>
> [2] Kim, Jun-Hyuk, Byeongho Heo, and Jong-Seok Lee. Joint Global and Local Hierarchical Priors for Learned Image Compression. In Proceedings of the IEEE/CVF Conference on Computer Vision and Pattern Recognition, pp. 5992-6001. 2022.
>
> [3] Dailan He, Ziming Yang, Weikun Peng, Rui Ma, Hongwei Qin, and Yan Wang. ELIC: Efficient learned image compression with unevenly grouped space-channel contextual adaptive coding. In Proceedings of the IEEE/CVF Conference on Computer Vision and Pattern Recognition, pp.5718–5727, 2022.

---

> > ### Author Response · Authors · 2022-11-11
> > **To Reviewer 7uFq (cont.)**
> >
> > > ``Significant of improvements''
> >
> > Fig.1(a) shows the improvements of our mask decay are significant. Previous work found reducing the model's capacity will result in a large RD performance loss. This crucial problem is firstly alleviated by our method. Compared to previous works, our method improves the Pareto frontier by a large margin. Note that relative improvement is more important for KD methods. And our mask decay improves our medium and small models by 50% and 30%, respectively.
> >
> > > ``The training strategy''
> >
> > The training strategy has been introduced on Page 5, and further explained in Appendix D. As stated on Page 5, ``we stop mask decay for one mask if it is sparse enough''. If the channels of teacher/student are 192/64, respectively, mask decay will be terminated when the mask has 64 positive and 128 zeros.

---

> > > ### Author Response · Authors · 2022-11-15
> > > **To Reviewer 7uFq (cont.)**
> > >
> > > As the deadline for discussion stage 1 is coming, we can only update our draft until Nov 18. If you have more suggestions, would you please provide them for us?

---

> > > > ### Comment · Reviewer_7uFq · 2022-11-16
> > > > **Response**
> > > >
> > > > Thanks for your response. The comparison between the proposed method and ELIC indicates the proposed method achieves very similar performance as the ELIC method. While the proposed method proposes a variable bitrate solution, it is not new enough since there are a lot of variable bitrate solutions for learned image codecs.

---

> > > > > ### Author Response · Authors · 2022-11-16
> > > > > **To Reviewer 7uFq**
> > > > >
> > > > > Thanks for your responses.
> > > > >
> > > > > We agree that our large model achieves similar performance as ELIC-sm and several variable bitrate solutions have been proposed for learned image codecs. However, it is challenging to incorporate them, which means making a single variable-bit-rate model achieve SOTA RD performance and fast run time. This challenge was also recognized by other reviewers. And we believe a single model with variable-bit-rate plays an important role in real-world applications.
> > > > >
> > > > > Besides this contribution, there are more contributions in our paper. Three models with different complexities are proposed to meet different latency requirements. Previous papers suffer from a large RD penalty by reducing the model capacity. This serious problem is firstly addressed by us. In particular, the mask decay with a novel sparsity loss is proposed to narrow the RD performance gap between these models. Another contribution is our residual representation learning which can enable encoding scalability. For neural codec, this further shortens the distance to practical usage.
> > > > >
> > > > > We believe our paper has enough novel ideas and contributions with promising impacts on the neural image compression field.

---

> > > > > > ### Author Response · Authors · 2022-11-18
> > > > > > **To Reviewer 7uFq**
> > > > > >
> > > > > > Dear reviewer, do you have any concerns on our responses or the revised paper? Please let us know and address them.

---

### Official Review · Reviewer_Y5zE · 2022-10-25

**Confidence:** 4
**Correctness:** 4
**Technical Novelty And Significance:** 3
**Empirical Novelty And Significance:** 3
**Recommendation:** 8

**Clarity, Quality, Novelty And Reproducibility:**

The proposed method is novel and effective. The supplementary provides the details that are helpful for reproducing the proposed method. The authors promised to release codes in the abstract.

**Strength And Weaknesses:**

Strength: The proposed method is novel and effective.

Weaknesses: The reviewer has some concerns about the experiments:

(a) The authors should compare the RD performance, speed, etc. of the proposed method with the SOTA neural image compression methods, e.g., Xie et al., Enhanced Invertible Encoding for Learned Image Compression. ACM MM 2021; Cheng et al., Learned Image Compression with Discretized Gaussian Mixture Likelihoods and Attention Modules. CVPR 2020, etc.

(b) The result of Tecnick should be added to Table 2.

(c) HEVC dataset is a video dataset, which is rarely used for image compression. Did the authors test their image compression methods on all frames of each video in the HEVC dataset? Instead, CLIC is a popular dataset for image compression.


**Summary Of The Paper:**

This paper proposes an efficient single-model variable-bit-rate network, which is able to run at 30 FPS with 768x512 input images and still outperforms VVC for RD performance. By further reducing both encoder and decoder complexities, the small model even achieves 30 FPS with 1920x1080 input images. To bridge the performance gap between different capacities models, the authors meticulously design the mask decay, which transforms the large model’s parameters into the small model automatically. And a novel sparsity regularization loss is proposed to mitigate the shortcomings of Lp regularization.

**Summary Of The Review:**

The proposed method is novel and effective. The reviewer recommends to accept the paper, however, the authors should address the questions regarding the experiments.

---

> ### Author Response · Authors · 2022-11-11
> **To Reviewer Y5zE**
>
> Thanks for your review and appreciation!
>
> > ``Compare with previous methods''
>
> Thank you for providing these two references [1,2]. Both of them are high-impact neural image codecs with high RD performance. We have included these two references and made comparisons in our paper. Cheng2020 [2] proposed attention modules and the Gaussian Mixture Model for the entropy model, while Xie2021 [1] proposed the invertible neural network as the backbone. All of their techniques are effective. But their models are equipped with the autoregressive module [3], which is GPU unfriendly. Our models do not use the autoregressive module, and therefore run much faster than these two models. For RD performance, our large model achieves -1.1% BD-Rate over VVC, while BD-Rates of Xie2021 and Cheng2020 are -0.54% and 3.89%, respectively. Hence, ours also outperforms these two models for RD performance. In particular, we use a single model to handle all rate points, while they trained separate models.
>
> [1] Yueqi Xie, Ka Leong Cheng, and Qifeng Chen. Enhanced invertible encoding for learned image compression. In Proceedings of the 29th ACM International Conference on Multimedia, pp. 162–170, 2021.
>
> [2] Zhengxue Cheng, Heming Sun, Masaru Takeuchi, and Jiro Katto. Learned image compression with discretized gaussian mixture likelihoods and attention modules. In Proceedings of the IEEE/CVF Conference on Computer Vision and Pattern Recognition, pp. 7939–7948, 2020.
>
> [3] David Minnen, Johannes Balle, and George D Toderici. Joint autoregressive and hierarchical priors for learned image compression. In Advances in Neural Information Processing Systems, volume 31, pp. 1–10, 2018.
>
> > ``Results of HEVC, CLIC, and Tecnick''
>
> Thanks for this suggestion. We tested our image codecs on HEVC with the only first frame of each video sequence. Kodak and HEVC are prevalent benchmarks for image and video codecs, respectively. Image compression models are always treated as the intra model in video codecs. To this end, we present results on HEVC to show the potential of our models in video codecs. We also would like to report results on CLIC and Tecnick. Table 9 summarizes them. Same as that in HEVC and Kodak, mask decay helps these models achieve superior performance on both CLIC and Tecnick.

---

> > ### Comment · Reviewer_Y5zE · 2022-11-16
> > **Review after response**
> >
> > Thanks for the response. I keep my original recommendation "8: accept, good paper".

---

### Official Review · Reviewer_5xdh · 2022-11-03

**Confidence:** 4
**Correctness:** 3
**Technical Novelty And Significance:** 3
**Empirical Novelty And Significance:** 3
**Recommendation:** 8

**Clarity, Quality, Novelty And Reproducibility:**

The paper is well written, and conducted rich ablation studies to sufficiently prove an effectiveness of each new building block.

The mask decay based knowledge distillation, the sparsity regularization loss specialized for this KD, and their CNN framework based on resnet and depth-wise convolution based encoder and encoder along with an adjustable quantization control design seem to be novel.

They plan to make the code available in public, so full details of their internal network design that is not described in the paper may also be available.

**Strength And Weaknesses:**

* Strength

Their CNN only framework is more efficient and GPU friendly, compared to recent Transformer based models, while the performance is on par with more complex SoTA neural image compression models.

With an adjustable quantization step, their model handles variable RD trade-offs through a single model unlike previous works.

They successfully enabled knowledge distillation on an image compression model through mask decay with a new sparsity loss that regularizes more effectively than Lp losses.

Their residual representation learning technique enables scalable encoder design with various complexities while maintaining an overall performance.


* Weakness

More visual results would better present their results, currently none in the main paper.

There are several points they insist as their technical contributions, which are rather scattered thus hard to get a clear picture. It would be better to clarify each more clearly in an overview.

Mask merging can be better clarified within the main paper. Also, does the Fig.4(a)-Pruned and the derivation in Eqs.(5-12) in Appendix B match?

Figure 7 seems to be an important study, but hard to read due to small size.

The intention of discussions on channel importance at p8 is rather confusing, considering their knowledge distillation mainly targets channel reduction, which was successful according to their experiments.

**Summary Of The Paper:**

This paper proposes an efficient CNN-based neural image compression algorithm. Their efficient CNN framework achieves a high speed while maintaining a high RD performance on par with SoTA works, and also can handle variable RD trade-offs through a single model. They also propose to effectively compress the large teacher model through knowledge distillation using a sparse channel mask technique along with a decaying sparsity regularization. During the mask decay based knowledge distillation, they further apply residual representation learning on encoder for higher scalability.

**Summary Of The Review:**

This paper proposes a pure CNN-based efficient neural image compression model and a knowledge distillation algorithm based on mask decay. The network is more efficient and GPU friendly while maintaining an on-par performance compared to recent Transformer based models. The evaluation and analysis have been made thoroughly enough, proving the effectiveness of the proposed algorithm properly. In conclusion, the proposed work successfully solved a hard challenge on efficiency-performance trade-offs in image compression.

---

> ### Author Response · Authors · 2022-11-11
> **To Reviewer 5xdh**
>
> Thanks for your review and appreciation!
>
> > ``Clarify each technical contribution''
>
> Thanks for this suggestion. This work really takes us lots of effort, therefore many techniques are included. First, we propose an efficient single-model variable-bit-rate framework with high rate-distortion performance (outperforming VVC). This denotes three important factors in practice: speed, footprint, and RD performance. To the best of our knowledge, most recent works consider at most two aspects, whereas our framework incorporates all of them. Second, we propose mask decay with a novel sparsity loss. These two techniques are effective to improve the student and we successfully enabled knowledge distillation on image compression models. Third, we advocate the scalable encoder. The core idea is learning residual representation (RRL). With RRL and mask decay, our scalable encoder is comparable to the teacher model. For the whole picture of our contributions, we summarize and clarify them at the end of Sec. 1 in our main paper.
>
> > ``Clarify mask merging''
>
> Thanks for this suggestion. We have updated Claim 1 in Appendix B for better clarification. This part shows how to merge the mask into the neighbor conv. layers. The derivations in Eqs. (5-12) exactly match the ``merging'' process in Fig.4(a).
>
> > ``The small size of Fig. 7''
>
> Thanks for this suggestion. We have redrawn these figures and put replications in the Appendix with large sizes. We believe it is easier to read this time.
>
> > ``Discussions on channel importance''
>
> This part discusses a hot topic in the field of model pruning. As we stated in the main paper, many works have studied this problem on semantic tasks. Recent work indicates that randomly pruning channels is good enough. In our paper, we achieve a similar conclusion on neural image compression. We emphasize reusing the teacher's parameters is really helpful. But it is not encouraged to study which channels are vital for the performance. For example, assume the layers of the teacher/student have 192/64 channels. It should take 64 channels from the 192 channels by pruning. Many choosing strategies can be proposed, such as choosing channels with maximum magnitudes. And some researchers believe that a better strategy will result in a better pruned model. But we propose avoidable mask decay to challenge this opinion. If one argues that some channels are important, we can avoid choosing these channels to achieve comparable performance. Overall, we provide these discussions for inspiring future works. It is easy to figure out ideas to choose ``unimportant'' channels to prune (as many works in semantic vision tasks), but it should be evaluated soundly.
>
> > ``More visual results''
>
> Thanks for this suggestion. We really want to put visualizations from the appendix to the main paper. But due to the limited pages, it is difficult to save some spaces in the main paper to achieve this. The most useful information about these visualizations is included in the main paper (page 8). Reconstructions of our different capacities models are consistent. And no extra obvious artifacts are caused by our mask decay. Our model is significantly faster than VTM.

---

### Author Response · Authors · 2022-11-11
**To All Reviewers**

Thanks to all reviewers! We want to summarize the strengths proposed by reviewers and clarify our contributions here.

1. Our efficient single-model variable-bit-rate network with high rate-distortion (RD) performance

Reviewer 5xdh appreciated our CNN-only framework. She/He pointed out that CNN is more efficient and GPU-friendly than recent Transformer based models, and our model is on par with these more complex SOTA models. Both Reviewer 5xdh and RU83 noticed we used an adjustable quantization step to handle variable RD trade-offs, different from most existing models that train several models for each RD trade-off. Reviewer RU83 stated that there is no previous model that incorporates variable bit rate encodings, variable encode compute, and faster run time. Reviewer 7uFq thought our method is useful for practical neural image compression and the inference speed of our model is encouraging.

2. Our mask decay with a novel sparsity loss

Reviewer RU83 pointed out that our mask decay addresses a significant problem for neural image compression. Previous papers found reducing the model's capacity suffers from a large RD penalty. Our mask decay provides a new approach to narrowing the performance gap. Reviewer 5xdh also appreciated we successfully enabled knowledge distillation (KD) on an image compression model. We also proposed a novel sparsity loss, and most reviewers appreciated its novelty and effectiveness.

3. Our scalable encoder

We further enabled encoding scalability and proposed residual representation learning (RRL) with mask decay to boost performance. Reviewer 7uFq appreciated the motivation for this design, while reviewer 5xdh pointed out that our scalable model maintains an overall performance.

Overall, most reviewers thought our paper is well-motivated and our proposed methods are novel and effective. Each new technique is evaluated sufficiently by our rich ablation studies. Reviewer 5xdh appreciated we successfully solved a hard challenge on efficiency-performance trade-offs in image compression. Reviewer RU83 thought our paper has enough novel ideas with promising impacts on the neural image compression subfield.

At last, we want to summarize all updates of our paper. First, we update our inference by including all papers mentioned by reviewers and discuss them in our main paper. Second, some figures (Figs. 7, 8, 15, 16, 18, and 19) are updated for easier reading and clarification. Third, more details about our mask decay are introduced in Appendix B. Fourth, different methods for achieving scalable encoders are discussed in Appendix C. Fifth, more experimental results are included in Appendix F, such as comparisons with other baselines, inference speed for megapixel/s, results on CLIC and Tecnick, etc.

---

### Decision · Program_Chairs · 2023-01-20

**Decision:**

Accept: poster

**Justification For Why Not Higher Score:**

This paper provides a solid contribution, but the ideas are not sufficiently novel to warrant a spotlight or oral.

**Justification For Why Not Lower Score:**

The paper proposed a model for real-time neural image compression that works well for different rate-distortion tradeoffs, and contains new elements. The method looks like it is practically useful, and it contains useful ideas for researchers and practitioners interested in learning-based image compression.


**Metareview: Summary, Strengths And Weaknesses:**

The paper proposed a model for real-time neural image compression that works well for different rate distortion tradeoffs.

Strength:
The model achieves high compression speeds, high performance, and can handle different rate-distrotion tradeoffs in a single model. In addition, the paper is well written and contains a variety of ablation studies.

Weaknesses:
Code is not yet available, and a few additional comparisons, e.g., to ELIC, would be useful. Those are not critical, however, the current experiments give a good idea about the merits of the proposed method.

**Note From Pc:**

if the above contains the word "oral" or "spotlight" please see: "oral" presentation means -> notable-top-5% and "spotlight" means -> notable-top-25%. As stated in our emails, we are disassociating presentation type from AC recommendations

**Summary Of Ac-Reviewer Meeting:**

R1 is supportive of publication (8) and asked for a few relatively minor expositional changes. The authors addressed those.

R2 finds the method `novel and effective', is supportive of publication (8), and asked for an additional comparison and clarification. The authors added an additional comparison and the corresponding references.

R3 notes that the method is useful and that the proposed solution is interesting, but is not supportive of publication (3). R3 argues that important baselines are missing (in particular ELIC, a recently proposed method), that transformers are not a good choice for the baselines methods, and finds the novelty lacking. The authors added some of the suggested additional comparisons in the rebuttal. The reviewer responded that those comparisons show that there are methods that have similar performance, and notes that the contributions regarding the variable bitrate is not sufficiently novel.

In my opinion, the paper provides a practical solution to image compression, and the proposed solution contains a variety of novel elements, and sufficiently many comparisons to the literature have been carried out to get a good idea about the merits of the method. Regarding the comparison to ELIC; ELIC is a very recent paper (July 2022), the authors implemented the method, but could not reproduce the results in the ELIC paper. Thus I don't think that the comparison is critical for the acceptance of the paper.

R4 is also supportive of publication. R4 made a few suggestions and also proposed to compare to ELIC, a recently proposed method.